# Personalized Vision via Visual In-context Learning

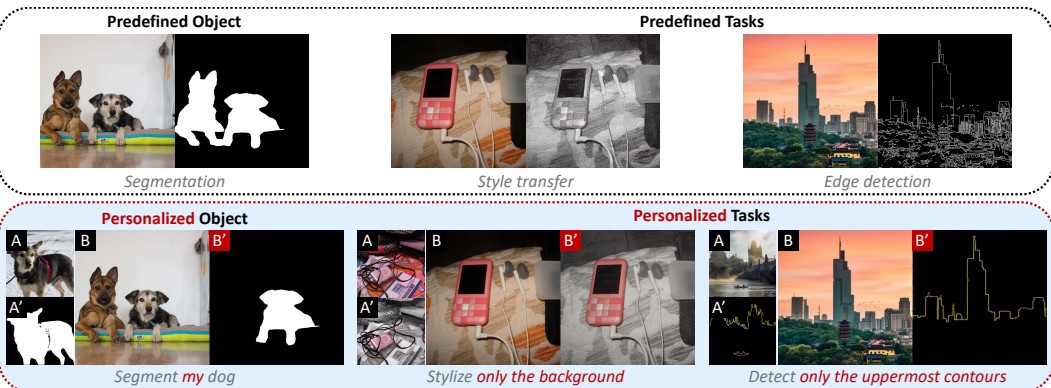

Figure 1: **Predefined vs. Personalized Vision.** Illustration of traditional vision tasks (top) and the personalized tasks enabled by our proposed **PICO** (bottom). Given a contextual example pair $(A \rightarrow A')$ defining the desired visual transformation, and a query image $B$, our model infers the task and generates the corresponding $B'$ at test time.

## Abstract

Modern vision models, trained on large-scale annotated datasets, excel at predefined tasks but struggle with personalized vision—tasks defined at test time by users with customized objects or novel objectives. Existing personalization approaches rely on costly fine-tuning or synthetic data pipelines, which are inflexible and restricted to fixed task formats. Visual in-context learning (ICL) offers a promising alternative, yet prior methods confine to narrow, in-domain tasks and fail to generalize to open-ended personalization. We introduce Personalized In-Context Operator (PICO), a simple four-panel framework that repurposes diffusion transformers as visual in-context learners. Given a single annotated exemplar, PICO infers the underlying transformation and applies it to new inputs without retraining. To enable this, we construct VisRel, a compact yet diverse tuning dataset, showing that task diversity, rather than scale, drives robust generalization. We further propose an attention-guided seed scorer that improves reliability via efficient inference scaling. Extensive experiments demonstrate that PICO (i) surpasses fine-tuning and synthetic-data baselines, (ii) flexibly adapts to novel user-defined tasks, and (iii) generalizes across both recognition and generation.

## 1 Introduction

Modern vision models (Radford et al., 2021; Oquab et al., 2024; Kirillov et al., 2023; Rombach et al., 2022; Esser et al., 2024), trained on large-scale annotated datasets, have achieved impressive performance in both visual recognition and generation. However, these models typically succeed on predefined object categories (*e.g.*, cars, people) or standard task formats (*e.g.*, object detection, semantic segmentation) where abundant labeled data exists. They often struggle to adapt flexibly to **personalized vision—tasks defined by users at test-time**, involving customized objects or novel task definitions. With growing demand for personalized vision systems that quickly adapt to individual needs, a critical question emerges: *How can we achieve flexible and high-performing personalized vision?*

A traditional approach to personalized vision uses generative models to synthesize additional training data tailored to specific personalized objects. For example, Personalized Representation (PRPG) (Sundaram et al., 2025) employs DreamBooth (Ruiz et al., 2023) to generate synthetic data for target concepts, then adapting general-purpose feature representations into personalized ones. While these methods (Sundaram et al., 2025; Zhang et al., 2024b) make strides toward personalized vision by adapting to personalized objects, they remain constrained to predefined task (*e.g.*, segmentation or classification) and require costly fine-tuning for each new subject. They do not generalize flexibly to arbitrary, user-defined tasks.

In natural language processing (NLP), in-context learning (ICL) (Brown et al., 2020; Dong et al., 2024) has shifted practice from task-specific fine-tuning toward models that can perform novel tasks defined at test time. A natural analogy in vision is to let exemplars define the task. However, unlike text, vision tasks have heterogeneous output format (*e.g.,* pixel arrays, masks, coordinates), making in-context generalization more challenging. Existing visual ICL methods (Bar et al., 2022; Wang et al., 2023a; Bai et al., 2024) unify tasks but fall short of personalized vision: they are typically evaluated on predefined, narrow, in-domain tasks and show limited generalization beyond training set, rather than adapting to open-ended personalized tasks at test time.

To address this gap, we study personalized task generalization: adapting to novel tasks or novel objects during test time. We introduce the **Personalized In-context Operator (PICO)**, a simple visual ICL framework based on a four-panel input format, where an annotated exemplar $(A \to A')$ defines the task and the model infers the underlying transformation and applies it to new inputs $(B \to B')$. To support this setting, we construct the **VisRel dataset**, a compact yet diverse tuning dataset of structurally organized visual tasks, designed to expose the model to a unified *visual-relation space* for broad generalization. To mitigate stochastic sampling variability, we propose an **attention-guided seed scorer** that leverages early-step cross-grid attention patterns to rank candidate seeds, enabling efficient test-time scaling.

We conduct extensive experiments to validate the effectiveness of PICO. First, PICO outperforms fine-tuning–based approaches on personalized subjects within conventional vision tasks. Second, PICO flexibly adapts to novel, user-defined tasks at test time, supported by both quantitative and qualitative results. Finally, PICO achieves strong performance across diverse personalized vision scenarios, covering both recognition and generation.

In summary, our key contributions are:

- We formulate **personalized vision as visual in-context learning**, enabling a single generative prior to adapt at test time to both new objects and new tasks from exemplars, without requiring synthetic data or costly per-subject fine-tuning.

- We construct the **VisRel dataset**, a compact yet diverse tuning dataset, and show that task diversity, rather than scale, drives strong generalization in visual ICL.

- We propose an **attention-guided seed scorer** that leverages early attention dynamics, improving the reliability stochastic generative sampling through efficient inference scaling.

- We demonstrate, across diverse benchmarks, that PICO achieves strong personalization performance with minimal supervision, spanning both recognition and generation, and covering varied subjects and task definitions.

## 2 RELATED WORK

**Personalized Vision.** Existing personalized vision methods (Sundaram et al., 2025; Zhang et al., 2024b; Alaluf et al., 2024; Cohen et al., 2022; Nguyen et al., 2024; Zhang et al., 2024a; Samuel et al., 2024) typically adapt vision or vision-language models (VLMs) to handle user-specific concepts within predefined tasks like retrieval and segmentation. For example, PerSAM (Zhang et al., 2024a) segments user-indicated regions using cosine similarity on pretrained segmentation features (Kirillov et al., 2023), while PDM (Samuel et al., 2024) leverages intermediate features from text-to-image (T2I) models (Rombach et al., 2022) to localize personalized instances. PRPG (Sundaram et al., 2025) generates synthetic training data to enhance personalized representations for downstream tasks. However, these methods are inherently restricted to fixed task formats, lacking flexibility to accommodate arbitrary user-defined tasks at test-time. Real-world personalization often

demands versatile, dynamically defined tasks (*e.g.*, inserting custom objects, generating annotations in new formats). Such scenarios motivate our approach to enable personalized vision systems to rapidly adapt beyond fixed frameworks.

**Visual In-Context Learning.** Visual ICL, inspired by prompt-based task adaptation in NLP (Brown et al., 2020), aims to adapt vision models to downstream tasks through contextual examples. Bar et al. (2022) first propose visual prompting by framing vision tasks as quad-grid masked image inpainting. Painter (Wang et al., 2023a), a ViT-based model (Dosovitskiy et al., 2021) trained through masked image modeling, shows strong ICL capabilities across various dense prediction tasks, and SegGPT (Wang et al., 2023b) further enhances this ability specifically for segmentation. However, these training-based visual ICL methods (Bar et al., 2022; Wang et al., 2023a;b) rely heavily on extensive, task-specific pretraining, limiting generalization to unseen tasks. In contrast, inference-based methods (Nguyen et al., 2023; Yang et al., 2023; Zhao et al., 2024; Gu et al., 2024; Lai et al., 2025) attempt to interpret visual demonstrations by translating them into textual instructions, which underuses visual signals and remains confined to semantic editing tasks, leading to inaccuracies from ambiguous text descriptions. Our work advances visual ICL by explicitly formulating personalized vision as visual relations within a unified space, enabling robust, flexible one-shot personalization tailored to individual needs.

**Diffusion Priors.** Diffusion models have emerged as the defacto paradigm for image synthesis (Rombach et al., 2022; Esser et al., 2024), demonstrating powerful generative priors beneficial for diverse vision tasks, including dense prediction (He et al., 2025b; Fu et al., 2024; Ke et al., 2024), image restoration (Xia et al., 2023; Wang et al., 2024; He et al., 2025a; Lugmayr et al., 2022), style transfer (Chung et al., 2024; Jiang et al., 2025), etc. Within data-scarce personalized vision settings, diffusion models are commonly employed to synthesize additional training data for downstream fine-tuning (Ruiz et al., 2023; Gal et al., 2023). However, this two-stage process (Sundaram et al., 2025) is computationally intensive, limiting practicality for frequent adaptation to personalized concepts. Recent work such as In-Context LoRA (Huang et al., 2024) have highlighted intrinsic ICL ability in diffusion transformers (Peebles & Xie, 2023). Building on these insights, we directly use diffusion priors as visual in-context learners, enabling flexible, immediate adaptation to arbitrary user-defined tasks without synthetic augmentation or retraining.

## 3 METHOD

Our objective is to achieve flexible visual personalization through a task-agnostic framework that adapts to user-defined tasks at inference, without additional fine-tuning. We reformulate personalized vision as a visual ICL problem, where a single input-output exemplar defines the task, and the model infers user intent from this demonstration and applies it to new queries. Central to our approach is learning a broad *visual-relation space*, repurposing pretrained diffusion transformers into in-context visual reasoners. We further introduce a lightweight seed-selection strategy for inference scaling that enhances stability and reliability.

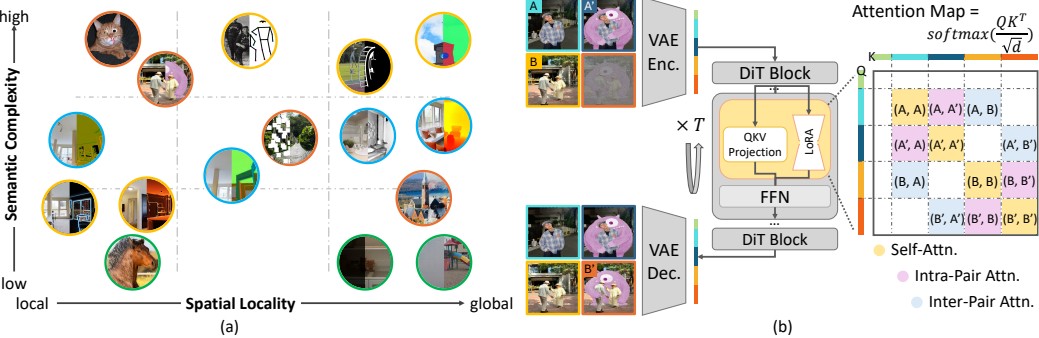

Figure 2: **(a) Structured Visual Relation Space.** Tasks are organized by semantic complexity (low to high) and spatial locality (local to global), covering diverse task types, color-coded as: ■ restoration/enhancement, ■ physical/geometric estimation, ■ semantic perception, ■ generative manipulation. **(b) Training pipeline of PICO.**

## 3.1 Data: A Visual Relation Space

ICL succeeds in NLP because every task (*e.g.*, translation, summarization, question answering, etc.) shares a unified language generation interface. In vision, however, different tasks have heterogeneous output format (*e.g.*, pixel arrays, masks, coordinates), limiting the potential for unified in-context generalization. We address this by unifying visual tasks as image-to-image transformations represented as RGB inputs and outputs (Bar et al., 2022; Wang et al., 2023a). Our key insight is that a robust visual ICL model should similarly embed tasks within a unified visual relation space, enabling interpolation and composition of transformations at test time. To learn this space, we curate VisRel, a compact yet diverse dataset of 27 visual tasks, aiming to span the space of common 2D transformations (see Figure 2(a)). Its design follows three principles.

**Task Taxonomy.** We structure the visual relation space along two intuitive axes: (1) *Semantic Complexity* measures the level of semantic understanding required, spanning low-level (pixel/color adjustments), mid-level (structure/shape manipulation), to high-level (object/class reasoning) transformations. (2) *Spatial Locality* defines the spatial context dependency, ranging from local (neighboring pixels), intermediate (objects patches), to global (full-image context) operations.

**Intra-task Diversity.** Each task includes diverse variants to avoid overfitting. For instance, inpainting uses masks of varying colors, shapes, and transparency; segmentation supports different colors, transparency mask; restoration tasks (denoising, deblurring) include multiple noise levels or blur kernels. This encourages learning transferable transformation principles rather than memorizing task-specific patterns, which is important for zero-shot generalization to novel personalized tasks.

**Minimal Text Label.** The model primarily relies on visual exemplars, but minimal text prompts help resolve ambiguities between potential conflicts of interest tasks (*e.g.*, local vs. global edits; black and white depth estimation vs. colorful style transfer). These lightweight cues (*e.g.*, "edit" vs. "estimate") act as soft boundaries while keeping the framework largely vision-driven.

## 3.2 Training: PICO

Given an exemplar pair $\{A, A'\}$ illustrating a visual relation $r : A \to A'$ and a query image $B$, the goal is to synthesize an output $B'$ that applies $r$ to $B$. We adopt a quad-grid input format

$$I = \text{GRID}\left(\begin{bmatrix} A & A' \\ B & B' \end{bmatrix}\right). \tag{1}$$

The training pipeline is illustrated in Figure 2(b). We build upon a pretrained T2I diffusion transformer (DiT) (Labs, 2024), finetuned with LoRA (Hu et al., 2022). A VAE encoder $\mathcal{E}(\cdot)$ maps the grid into latent space, yielding the target $x_0 = \mathcal{E}(B')$ and visual conditions $c_{\text{vp}} = \mathcal{E}(\{A, A', B\})$, while $c_{\text{txt}}$ encodes minimal text prompts. At time $t$, the latent sequence is $Z_t = [\, x_t; \ c_{\text{vp}}; \ c_{\text{txt}}\,]$, where the target latent is noised as $x_t = (1 - t)x_0 + t\,\epsilon$, with $\epsilon \sim \mathcal{N}(0, I)$ and $t \sim \mathcal{U}(0, 1)$. Each DiT block applies multi-modal attention. For head $h$ in block $b$,

$$\text{MMA}^{(b,h)}(Z_t) = \text{softmax}\left(\frac{Q_t^{(b,h)} K_t^{(b,h)\top}}{\sqrt{d_h}}\right) V_t^{(b,h)}, \tag{2}$$

where $Q_t, K_t, V_t$ are projections of $Z_t$, $H$ is the number of heads.

**Clean noising and Objective.** Unlike In-Context LoRA (Huang et al., 2024), which perturbs all latents, we inject noise solely into the target $x_0$, leaving $c_{\text{vp}}, c_{\text{txt}}$ clean. This prevents corruption of exemplar information and yields stable relation transfer. The training objective is applied only on the target quadrant, so the model focuses on reconstructing $B'$ while leveraging the clean context for guidance. Concretely, the model predicts a conditional velocity field $v = v_\Theta(x_t, t \mid c_{\text{txt}}, c_{\text{vp}})$, trained with conditional flow matching (CFM):

$$\mathcal{L}_{\text{CFM}} = \mathbb{E}_{t, x_t}\left[\|v_\Theta(x_t, t \mid c_{\text{txt}}, c_{\text{vp}}) - \hat{v}(x_t, t)\|^2\right], \tag{3}$$

where $\hat{v}(\cdot)$ is the oracle velocity defined by the flow schedule.

## 3.3 Inference: One-shot personalization.

At test time, the $B'$ quadrant is replaced by a placeholder $X$, initialized as Gaussian noise in latent space. The three context quadrants remain clean: $c_{\text{vp}} = \mathcal{E}([A, A', B])$. Starting from $x_1 \sim \mathcal{N}(0, I)$,

we integrate the learned flow

$$\frac{dx_t}{dt} = v_\Theta(x_t, t \mid c_{\text{txt}}, c_{\text{vp}}),$$

(4)

from $t = 1 \rightarrow 0$ to obtain $x_0$, and decode $B' = \mathcal{D}(x_0)$, where $\mathcal{D}$ is the VAE decoder. The model seamlessly transfers the visual transformation demonstrated by $(A, A')$ to the query $B$, supporting flexible, test-time personalization without fine-tuning.

**Inference scaling via Attention-Guided Seed Selection.** Generative sampling is stochastic: different seeds can diverge, which is undesirable for deterministic or localized tasks (Figure 8). We introduce a training-free seed scorer that exploits early cross-attention routing (Eq. 2) to select promising seeds before full denoising. We refer to the bottom-right (BR) quadrant as the target region. During training, BR contains the ground-truth $B'$; during inference, BR is the placeholder $X$. Our intuition is that BR queries should initially bind to evidence in $B$ and transformation cues in $(A, A')$, then pivot back to BR; persistent focus on exemplars risks copying rather than adapting.

Let $p_{s,b,i}^{\text{br}}$ and $p_{s,b,i}^{\text{vp}}$ denote the average attention mass from target BR queries to BR keys and to visual context keys $(A, A', B)$, respectively, at early solver step $i$ for seed $s$ (averaged over heads $H$). For blocks $\mathcal{B}^\dagger$ and the first few solver steps $i \in \{0, 1, 2\}$, we measure the pivot:

$$D_{\text{br}}(s) = \frac{1}{|\mathcal{B}^\dagger|} \sum_{b \in \mathcal{B}^\dagger} \left(p_{s,b,2}^{\text{br}} - p_{s,b,0}^{\text{br}}\right), \quad D_{\text{vp}}(s) = \frac{1}{|\mathcal{B}^\dagger|} \sum_{b \in \mathcal{B}^\dagger} \left(p_{s,b,2}^{\text{vp}} - p_{s,b,0}^{\text{vp}}\right).$$

(5)

Here, $D_{\text{br}}(s)$ quantifies how strongly queries pivot toward the target, while $D_{\text{vp}}(s)$ captures how quickly they peel away from exemplars. The final seed score is

$$S_{\text{pivot}}(s) = z(D_{\text{br}}(s)) - z(D_{\text{vp}}(s)), \quad s^\star = \arg\max_{s \in \mathcal{S}} S_{\text{pivot}}(s),$$

(6)

with $z(\cdot)$ denoting $z$-normalization across candidate seeds $\mathcal{S}$. Pseudo-code and further statistical analysis are in the Appendix B.

## 4 EXPERIMENTS

We validate our method through extensive experiments addressing three key questions: (1) Does visual ICL surpass traditional personalized fine-tuning on standard tasks like personalized segmentation? (2) Can the framework handle novel, user-defined tasks at inference? (3) Does it extend across recognition and generation tasks?

### 4.1 IMPLEMENTATION DETAILS

We build PICO upon FLUX.1-dev (Labs, 2024), a latent rectified flow transformer model, finetuning with LoRA (Hu et al., 2022) (rank 256) on the VisRel dataset for $30,000$ steps using a single H100 GPU. All experiments are conducted at a resolution of $1024 \times 1024$, with each cell of the quad-grid structured as $512 \times 512$. We use the Prodigy optimizer (Mishchenko & Defazio, 2024) with safeguard warmup, bias correction enabled, and a weight decay of $0.01$. The VisRel training dataset contains $315$ samples across $27$ diverse tasks, curated from existing sources. Details of data construction are provided in Appendix A, and for transparency we include a one-page contact sheet of all images in the supplementary material. For fair comparison, we report results using a single default seed. Results annotated with TTS additionally employ our proposed test-time scaling strategy. In this setting, we fix $\mathcal{B}^\dagger = \{9, 11, 12\}$, probe steps $\{0, 1, 2\}$, and use a candidate seed set of size $|\mathcal{S}| = 10$. Code and model will be released.

### 4.2 PERSONALIZED IMAGE SEGMENTATION

**Datasets.** We evaluate across four personalized segmentation benchmarks: PerSeg (Zhang et al., 2024a), DOGS (Sundaram et al., 2025), PODS (Sundaram et al., 2025), and PerMIS (Samuel et al., 2024). While PerSeg and DOGS mainly contain either single instances or distinct instances easily segmented using semantic cues, PODS is more challenging due to variations in viewpoints, scales, and distractors. PerMIS, sourced from video frames, further increases the difficulty by emphasizing instance-level segmentation.

Table 1: **Quantitative Comparison on personalized segmentation.** We compare PICO with diverse baselines. ★: best, ☆: second-best, and ◆: third-best.

| Method | PerSeg | | | DOGS | | | PODS | | | PerMIS | | |
|---|---|---|---|---|---|---|---|---|---|---|---|---|
| | mIOU↑ | bIOU↑ | F1↑ | mIOU↑ | bIOU↑ | F1↑ | mIOU↑ | bIOU↑ | F1↑ | mIOU↑ | bIOU↑ | F1↑ |
| *large-scale* | | | | | | | | | | | | |
| PerSAM | 90.50◆ | 72.79◆ | 94.07☆ | 86.87☆ | 71.06★ | 53.18 | 67.45☆ | 56.63☆ | 45.60★ | 51.77☆ | 37.95☆ | 21.71☆ |
| SegGPT | 95.77★ | 81.58★ | 99.16★ | 91.16★ | 65.93☆ | 85.14★ | 65.22★ | 50.75★ | 42.45 | 77.90★ | 47.10★ | 38.61★ |
| *personalized* | | | | | | | | | | | | |
| PDM | 29.99 | 10.97 | 2.79 | 21.03 | 8.95 | 0.11 | 26.39 | 10.98 | 1.12 | 23.62 | 9.10 | 1.27 |
| PDM+PerSAM | 50.09 | 60.08 | 33.37 | 64.36 | 53.82 | 41.85 | 35.56 | 45.34 | 22.33 | 28.93 | 25.25 | 11.72 |
| PRPG | - | - | - | 81.52◆ | 37.34 | 68.74☆ | 60.68 | 34.56 | 40.41◆ | - | - | - |
| *generalist* | | | | | | | | | | | | |
| VP | 24.83 | 18.11 | 0.03 | 38.50 | 14.34 | 4.86 | 17.48 | 12.10 | 0.14 | 8.87 | 4.16 | 0.10 |
| Painter | 56.56 | 51.58 | 29.76 | 72.07 | 49.75 | 56.88◆ | 26.93 | 25.44 | 6.87 | 19.53 | 15.59 | 4.20 |
| LVM | 43.86 | 33.92 | 19.49 | 54.65 | 27.96 | 30.23 | 22.64 | 13.50 | 2.00 | 16.38 | 8.73 | 1.14 |
| OmniGen | 33.24 | 37.33 | 9.52 | 44.87 | 41.48 | 18.54 | 20.75 | 20.57 | 2.19 | 13.43 | 14.88 | 1.77 |
| PICO (ours) | 90.97☆ | 76.13◆ | 62.82◆ | 71.02 | 54.71◆ | 49.84 | 68.72★ | 60.26★ | 44.88☆ | 49.52◆ | 33.63◆ | 14.90◆ |
| PICO+TTS | 92.04 | 76.85 | 98.14 | 72.33 | 56.54 | 59.62 | 69.90 | 63.60 | 48.50 | 50.66 | 35.23 | 15.96 |
| Δ w/ TTS | +1.07 | +0.12 | +35.32 | +1.31 | +1.83 | +9.87 | +1.18 | +3.34 | +3.62 | +1.14 | +1.60 | +1.06 |

Table 2: **Comparison of baseline methods.** (Top) personalized segmentation methods; (Bottom) other methods. PICO uses minimal supervision with a diffusion backbone and remains flexible for novel test-time tasks.

| Method | Use of Generative Prior | Features | Seg. Method | Test-time New Instance? |
|---|---|---|---|---|
| PDM | Feature extractor | SDXL-turbo (Sauer et al., 2024) | Attention map | ✓ |
| PRPG | Synthetic data generator | Personalized DINOv2 (Oquab et al., 2024) | Attention map | ✗ (retraining required) |
| PICO (ours) | In-context learner | – | Direct output | ✓ |

| Method | Seg. Data / Total Data | Training | Loss |
|---|---|---|---|
| PerSAM | 11M / 11M | Finetune from MAE-pretrained ViT-H (He et al., 2022) | Cross-entropy |
| SegGPT | 254K / 254K | Finetune from Painter (Wang et al., 2023a) | Smooth L1 |
| VP | - / 88K unlabeled Arxiv Data | Finetune from MAE-VQGAN (He et al., 2022; Esser et al., 2021) | Cross-entropy |
| Painter | 138K / 192K | Finetune from MAE-pretrained ViT-Large (He et al., 2022) | Smooth L1 |
| LVM | 10.1M / 1.68B | Train LLaMA-style (Touvron et al., 2023) transformer from scratch | Cross-entropy |
| OmniGen | 313K / 0.1B | Finetune from Phi-3 (Abdin et al., 2024) | Flow-matching |
| PICO (ours) | 40 / 315 | Finetuned from FLUX (DiT-based) (Labs, 2024) | Flow-matching |

**Baselines.** We compare PICO with three groups of state-of-the-art methods: (i) Large-scale pretrained segmentors: PerSAM (Zhang et al., 2024a) and SegGPT (Wang et al., 2023b), both trained on extensive collections of annotation segmentation masks. (ii) Personalized representation learners: PDM (diffusion features) (Samuel et al., 2024) and PRPG (personalized features via synthetic-data finetuning) (Sundaram et al., 2025), followed by using attention maps for instance localization. (iii) Generalist ICL models: Visual Prompting (VP) (Bar et al., 2022), Painter (Wang et al., 2023a), LVM (Bai et al., 2024) and OmniGen (Xiao et al., 2025).

**Evaluation Metrics.** Following (Samuel et al., 2024; Sundaram et al., 2025), we report mIOU, bIOU and F1@0.50 scores over all benchmarks. All the baseline methods we use its official code base and default settings.

**Results.** Table 1 shows that PICO outperforms generalist ICL models (VP, Painter, LVM, OmniGen) and personalized representation methods (PDM, PRPG), particularly on the more challenging PODS and PerMIS datasets. While PRPG achieves competitive results on DOGS, its reliance on per-instance synthetic data generation makes it computationally costly and difficult to scale (see Table 2(Top)). Thus, we omit its results on PerSeg and PerMIS, where over 500 unique instances are each accompanied by a single reference image. In contrast, PICO's generative in-context learning paradigm enables instant adaptation to new instances at inference without retraining, offering strong practical advantages. Notably, compared to large-scale pretrained segmentors, PICO achieves comparable performance while using up to four orders of magnitude fewer labeled data (see Table 2(Bottom)), highlighting its superior data efficiency enabled by generative priors. Qualitative results are provided in Figure 19 of the Appendix.

**Free-Form Inputs and Task Flexibility.** Beyond dense masks, PICO supports free-form inputs such as sparse annotations (*e.g.*, bounding boxes, circles) or part-level references, offering greater flexibility for personalization. As shown in Figure 3, PICO generates diverse segmentation outputs conditioned on visual exemplars while keeping text prompts fixed. Outputs vary along multiple

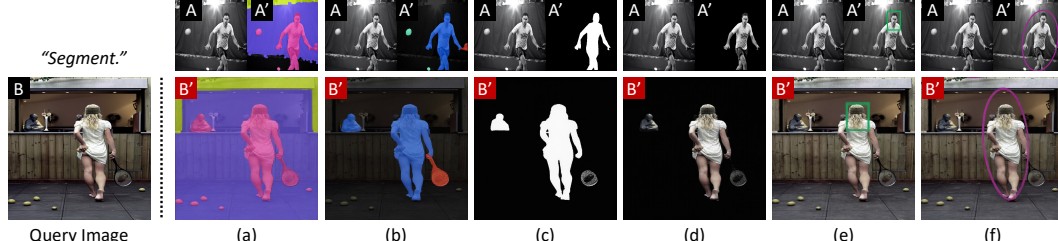

Figure 3: **Personalized segmentation with visual prompt control.** Given the same query image $B$ and text prompt ("Segment"), PICO produces diverse outputs on $B$ by varying the visual exemplar $(A \to A')$, controlling task type, style, granularity, and spatial focus.

dimensions: (i) *Task type*: stuff (a) vs. semantic (b) segmentation with arbitrary color coding and transparency; (ii) *Style*: binary silhouettes (c) vs. matting-like masks (d); (iii) *Granularity*: dense masks vs. sparse annotations; (iv) *Spatial focus*: whole-object (f) vs. part-level regions (e). PICO reliably aligns with the intent, style, and semantics conveyed in visual prompts, which are often hard to specify in text. Additional analyses of visual prompt effects are in Appendix C.1.

## 4.3 PERSONALIZED TEST-TIME TASK GENERALIZATION

**Task Definition.** We evaluate test-time personalization on user-defined visual tasks that differ from conventional CV setups. Specifically, we focus on: (i) **Composite tasks** requiring multi-step operations (*e.g.*, watermark removal followed by stylization). (ii) **Spatially constrained tasks**, traditionally performed globally but here applied locally or selectively (*e.g.*, contour-only edge detection, background-only stylization). (iii) **Semantic-conditional tasks** demanding context-aware edits (*e.g.*, adding stickers to semantically relevant image regions).

**Baselines.** Given these novel tasks, we compare PICO with representative state-of-the-art methods supporting visual instructions, including: (i) Inference-based method: VP (Bar et al., 2022), Analogist (Gu et al., 2024); (ii) Training-based method: PromptDiffusion (Wang et al., 2023c), LVM (Bai et al., 2024), OmniGen (Xiao et al., 2025), InstaManip (Lai et al., 2025); (iii) *Commercial multimodal models*: GPT-4o (OpenAI, 2024). Textual instructions for these methods follow Analogist's GPT-4o-based reasoning procedure.

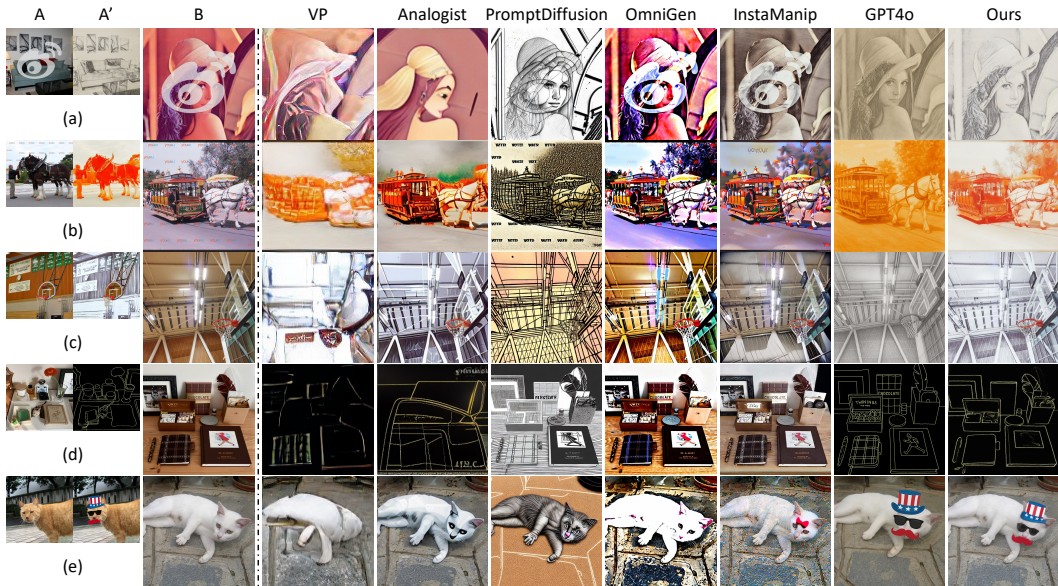

Figure 4: **Qualitative comparisons on test-time personalized tasks.** Each task is defined by a visual exemplar $(A \to A')$. We compare PICO with five representative baselines on: (a)(b) watermark removal + style transfer; (c) background-only stylization; (d) contour-only edge detection; and (e) sticker insertion.

Table 3: **Quantitative comparison on test-time personalized tasks.** The best results are in **bold**, second-best are underlined. GPT-4o* results are based on 10 random samples due to API constraints.

| | Ref | OmniGen | LVM | VP | Analogist | PromptDiff | InstaManip | GPT-4o* | PICO (Ours) |
|---|---|---|---|---|---|---|---|---|---|
| *deraining with inpainting* | | | | | | | | | |
| PSNR (dB)↑ | ∞ | 15.63 | 15.39 | 14.62 | 12.35 | 9.64 | 10.94 | 12.29 | **22.24** |
| SSIM ↑ | 1.0 | 0.47 | 0.35 | 0.36 | 0.35 | 0.10 | 0.33 | 0.26 | **0.67** |
| *inpainting with stylization* | | | | | | | | | |
| Gram↓ | 17.29 | 90.78 | 27.11 | 28.96 | 26.53 | 61.61 | 44.39 | 22.04 | **21.27** |
| FID↓ | 1.71 | 1.92 | 1.90 | 1.86 | **1.82** | 1.86 | 1.88 | 1.87 | 1.87 |
| LPIPS↓ | 0.62 | 0.59 | 0.61 | 0.82 | 0.70 | 0.77 | 0.60 | 0.68 | **0.52** |
| ArtFID↓ | 4.38 | 4.63 | 4.68 | 5.19 | 4.79 | 5.06 | 4.59 | 4.81 | **4.38** |

Table 4: **Quantitative results on w/wo texts.**

| Method | Pers. Seg↑ | Normal↓ | Z-depth↓ |
|---|---|---|---|
| VTM (10-Shot) | – | 11.4391 | **0.0316** |
| Ours w/o Text | 66.88 | 12.7105 | 0.0432 |
| Ours w Text | **68.72** | **10.5306** | 0.0377 |

| Method | 2DEdge↓ | 2DKeypoint↓ | Reshading↓ |
|---|---|---|---|
| VTM (10-Shot) | 0.0791 | 0.0639 | **0.1089** |
| Ours w/o Text | 0.0538 | 0.0609 | 0.1518 |
| Ours w Text | **0.0515** | **0.0497** | 0.1364 |

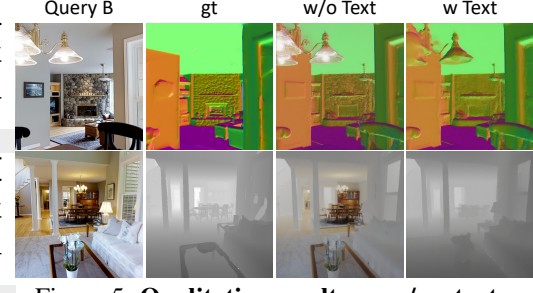

Query B    gt    w/o Text    w Text

Figure 5: **Qualitative results on w/wo texts.**

**Evaluation Metrics** We evaluate two representative composite tasks with clear quantitative protocols: (1) Deraining with inpainting, measured by PSNR and SSIM against clean ground truth. (2) Inpainting with stylization, measured by Gram distance for style fidelity, LPIPS for content preservation, and ArtFID (Chung et al., 2024) for overall perceptual quality. Full protocols and dataset details are provided in the Appendix D.1.

**Results.** Figure 4 shows that PICO effectively handles diverse test-time defined novel tasks, clearly surpassing all baselines. Training-based methods (PromptDiffusion, OmniGen, InstaManip) primarily target semantic-driven editing and struggle to match exemplar appearances, especially for non-RGB outputs (*e.g.*, edge maps in Figure 4(d)). Inference-based methods (VP, Analogist) can mimic target transformations roughly, but suffer from low fidelity and noticeable visual artifacts. GPT-4o (OpenAI, 2024) shows promising in-context reasoning, but two major limitations are observed: (1) Spatial misalignment: While semantic content is preserved, pixel layouts are distorted, harming precision tasks (see Figure 4(d-e)). (2) Over-reliance on abstract semantics: outputs rely on abstract semantics rather than exemplar fidelity, producing generic effect ( "sketch" or "orange-tone") in stylization tasks (see Figure 4(a-c)). In contrast, PICO produces outputs consistently aligned with exemplar cues in both spatial detail and semantic fidelity, demonstrating robust visual reasoning. Table 3 confirms this quantitatively, with PICO consistently achieving the best results across both tasks and metrics. Additional qualitative comparisons are provided in Appendix D.2.

## 4.4 ABLATION STUDIES

**Effects of Text Prompts.** We first quantify the importance of minimal textual prompts in disambiguating multiple visual tasks. Specifically, we evaluate on personalized segmentation (PODS) and five dense prediction tasks from Taskonomy (Zamir et al., 2018), using 1,000 test samples per task. Metrics follow (Kim et al., 2023): mean error (mErr) for surface normal, and RMSE for others. Predictions are converted from RGB to raw output space before scoring.

Table 4 shows that adding text prompts consistently improves performance, acting as soft task boundaries that reduce ambiguity beyond visual prompts alone. Figure 5 illustrates typical confusions without text, such as RGB-like outputs instead of surface normal maps. With text, the model cleanly separates task outputs. For reference, we include VTM (Kim et al., 2023), a state-of-the-art 10-shot fine-tuning method for dense prediction. Remarkably, our generative in-context learner surpasses this specialized approach on tasks such as surface normal estimation and texture edge detection, highlighting strong generalization and data efficiency enabled by generative priors.

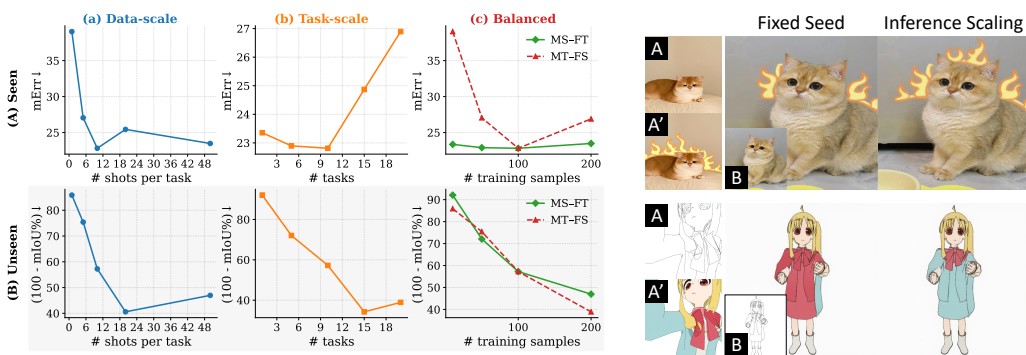

Figure 6: **Quantitative comparisons of different scaling strategies.** Lower values indicate better performance.

Figure 7: **Test-time scaling.** Results w/wo our attn-guided seed scorer.

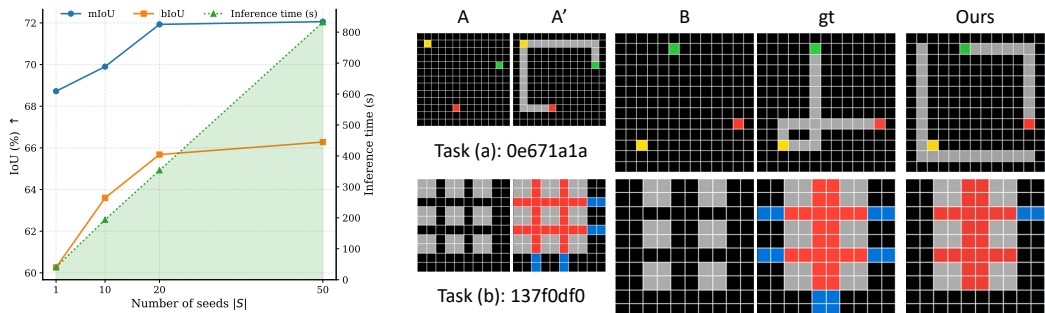

Figure 8: Effect of number of seeds on inference time and performance.

Figure 9: Qualitative examples of failure cases on ARC tasks that are truly OOD.

**Task vs. Data Scaling.** We systematically study how task diversity and data volume shape model generalization. With LoRA rank fixed ($r=128$) and $10k$ training steps, we evaluate three settings: (i) **Data-scale sweep:** fix 10 tasks, vary shots per task: ($K \in 1, 5, 10, 20, 50$). (ii) **Task-scale sweep:** fix 10 shots, vary number of tasks ($N \in 1, 5, 10, 15, 20$). (iii) **Balanced sweep:** fix total training images (10, 50, 100, 200), compare many-tasks–few-shots ($N > K$) against few-tasks–many-shots ($N < K$) regimes. We evaluate on both in-domain tasks seen during training (*e.g.*, surface normal estimation) and out-of-domain tasks not seen during training (*e.g.*, personalized segmentation).

Results are shown in Figure 6. For in-domain tasks, more data volume consistently improves performance (Figure.6A-a), while adding tasks hurts (Figure.6A-b), indicating limited capacity for memorizing multiple tasks. For out-of-domain generalization, performance improves with more data per task only up to 20 shots, after which it declines due to over-specialization (Figure.6B-a). In contrast, task diversity consistently boosts generalization (Figure.6B-b). Under fixed budgets, the many-tasks–few-shots strategy increasingly outperforms fewer-tasks–many-shots as task count grows (Figure.6B-c). These results support our *visual-relation–space* hypothesis: **data scaling helps memorization of seen tasks, whereas task diversity is key for robust generalization to novel, user-defined tasks.**

**Test-Time Scaling.** We evaluate our early-step seed-scoring strategy on personalized segmentation tasks and observe consistent improvements across all dataset (see Table 1). Qualitative examples in Figure 8 further show that scaled outputs align more faithfully with visual exemplars. We also report the computational overhead of the proposed TTS method in Figure 8. Inference time increases approximately linearly with the number of seeds $|S|$. Together with the performance curve, this reveals clear diminishing returns beyond $|S|=20$, considering the inference cost, we find $|S|=10$ to be a good trade-off. Additional ablations are provided in Appendix C.

## 5 CONCLUSION

In this paper, we present PICO, a novel approach for personalized vision by reformulating it as a visual in-context learning (ICL) problem. Unlike existing methods that rely heavily on task-specific

fine-tuning or synthetic data augmentation, PICO leverages a unified visual-relation space, enabling pretrained diffusion models to interpret user-defined tasks from a single visual demonstration at inference. Extensive experiments show that PICO adapts flexibly to novel objects and tasks, achieving strong performance across recognition and generation, and highlighting the potential of generative models as versatile visual in-context reasoners.

**Limitation and Future Work.** PICO generalizes well within the trained visual-relation space but is less reliable on entirely novel task types outside it. As illustrated in Figure 9, the model can produce outputs that appear visually plausible yet are logically incorrect. This aligns with human learning, *i.e.*, people extrapolate best within familiar domains, but broadening the method to truly novel tasks remains an open challenge. The four-panel input format, while effective, inherently limits the number and richness of demonstrations. Future work includes extending PICO to richer or sequential context (*e.g.*, videos or long-context models (Gu et al., 2025)) to broaden task coverage and strengthen visual reasoning.

## CODE OF ETHICS

The authors have read and acknowledge adherence to the ICLR Code of Ethics.

## ETHICS STATEMENT

All datasets used in this work are publicly available, widely adopted in the research community. We comply with dataset licenses and usage guidelines. Our curated VisRel dataset (Section A) is derived entirely from existing open-source benchmarks commonly used in vision research. Our experiments include both object-centric datasets and standard benchmarks that contain human figures (*e.g.*, COCO (Lin et al., 2014), CelebAMask-HQ (Lee et al., 2020)). Human figures appear only as part of these existing benchmarks to evaluate generalization across diverse visual domains. No private or newly collected human data was used.

Our contributions aim to advance personalized vision by enabling models that generalize to user-defined tasks during test time without requiring central sharing of user data, thereby reducing privacy risks compared to conventional methods.

A potential risk, as with much vision research, is misuse for surveillance applications. We explicitly focus on object-centric and task-based personalization, and do not target surveillance-related applications.

We also acknowledge the environmental impact of GPU training and inference. Our approach is comparatively data- and compute-efficient, requiring only a single GPU and modest training steps (Section 4), thereby limiting carbon footprint relative to prior work.

## REPRODUCIBILITY STATEMENT

We have taken deliberate steps to ensure reproducibility. The main paper (Section 4) specifies the model architecture, training objectives, evaluation metrics, and hyperparameter settings. The appendix provides dataset preparation details (Section A), and extended results. We justify each design choice with extensive ablations and full results for transparency. To further aid reproducibility, we include a visual overview of the training dataset: all 315 images are presented as a one-page contact sheet in the supplementary materials. We will release the code, the pretrained model and the dataset upon acceptance. Together, these measures ensure that researchers can replicate and extend our results without ambiguity.

## USE OF LARGE LANGUAGE MODELS

We used large language models (LLMs), such as GPT-5, only as general-purpose assistive tools to improve readability and refine the presentation of the paper. They did not contribute to research ideation, algorithm design, or experimental results. All technical content was independently conceived, implemented, and verified by the authors. This guarantees the scientific integrity and originality of this work.

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

APPENDIX

This appendix provides supplementary information not included in the main paper due to space constraints. Specifically, it includes details of the Visual Relation Dataset (VisRel) (Section A), additional explanations of test-time scaling (Section B), further ablation studies (Section C), extended analysis of test-time task generalization (Section D), and additional results (Section E).

## A    VISUAL RELATION DATASET (VISREL) DETAILS

The Visual Relation Dataset (VisRel) is a diverse collection of 2D visual tasks reformulated as visual transformations $(A \rightarrow A')$. It spans a wide range of task types and annotation formats, enabling the modeling of a unified visual relation space. It aims to trigger cross-task generalization and test-time adaptation via relation-space interpolation. VisRel integrates heterogeneous datasets, each of which contributing different visual relations. For clarity, we categorize them into four groups based on their underlying task semantics: (1) Image Restoration and Enhancement, (2) Physical and Geometric Perception, (3) Semantic Perception, and (4) Generative Manipulation.

Table 5 provides a detailed overview of the datasets included in VisRel. For each dataset, we list the task type, the visual transformation (input-output pair) that defines the task, and the annotation source. This diverse and well-structured dataset provides the foundation for our visual in-context learning framework, enabling PICO to generalize to novel user-personalized visual transformations at test time.

Table 5: **Summary of datasets in VisRel.** Each dataset is represented by its task type, exemplar relation $(A \rightarrow A')$, and annotation source. **Ground Truth** denotes annotations provided by the original dataset, while **Human-labeled** indicates annotations created by us.

| Dataset | Task Type | Visual Relation $(A \rightarrow A')$ | Annotation Source |
|---|---|---|---|
| **Restoration / Enhancement** | | | |
| DIV2K (Agustsson & Timofte, 2017) | Deblurring | Blurry Image → Clean Image | Ground Truth |
| DIV2K (Agustsson & Timofte, 2017) | Denoising | Noisy Image → Clean Image | Ground Truth |
| Synthetic Rain (Jiang et al., 2020) | Deraining | Rainy Image → Clean Image | Ground Truth |
| Dense-Haze Ancuti et al. (2019) | Dehazing | Hazy Image → Clean Image | Ground Truth |
| LOL (Wei et al., 2018) | Low-Light Enhancement | Low-Light Image → Enhanced Image | Ground Truth |
| **Physical / Geometric Perception** | | | |
| Taskonomy (Zamir et al., 2018) | Surface Normal Estimation | RGB Image → Surface Normal Map | Ground Truth |
| Taskonomy (Zamir et al., 2018) | Euclidean Distance Estimation | RGB Image → Distance Map | Ground Truth |
| Taskonomy (Zamir et al., 2018) | Z-buffer Depth Estimation | RGB Image → Z-buffer Map | Ground Truth |
| Taskonomy (Zamir et al., 2018) | Principal Curvature Estimation | RGB Image → Curvature Map | Ground Truth |
| Taskonomy (Zamir et al., 2018) | Reshading | RGB Image → Re-rendered Image | Ground Truth |
| Taskonomy (Zamir et al., 2018) | 2D Keypoint Estimation | RGB Image → 2D Keypoint Heatmap | Ground Truth |
| Taskonomy (Zamir et al., 2018) | 3D Keypoint Estimation | RGB Image → 3D Keypoint Heatmap | Ground Truth |
| Taskonomy (Zamir et al., 2018) | Occlusion Edge Detection | RGB Image → Occlusion Edge Map | Ground Truth |
| Taskonomy (Zamir et al., 2018) | Texture Edge Detection | RGB Image → Texture Edge Map | Ground Truth |
| **Semantic Perception** | | | |
| MS-COCO (Lin et al., 2014) | Instance Segmentation | Image → Instance Masks | Ground Truth |
| MS-COCO (Lin et al., 2014) | Panoptic Segmentation | Image → Panoptic Masks | Ground Truth |
| MS-COCO (Lin et al., 2014) | Semantic Segmentation | Image → Class Masks | Ground Truth |
| DIS5K (Qin et al., 2022) | Dichotomous Segmentation | Image → Binary Mask | Ground Truth |
| CORe50 (Lomonaco & Maltoni, 2017) | Object Detection | Image → Bounding Boxes | Human-labeled |
| MS-COCO (Lin et al., 2014) | Human Pose Estimation | Image → Keypoint Map | Ground Truth |
| OPRA (Fang et al., 2018) | Accordance Detection | Image → Highlighted Accordance Part | Ground Truth |
| **Generative Manipulation** | | | |
| DIV2K (Agustsson & Timofte, 2017) | Inpainting | Masked Image → Completed Image | Ground Truth |
| MS-COCO (Lin et al., 2014) | Style Transfer | Image → Stylized Image | Chung et al. (2024) |
| PhotoDoodle (Huang et al., 2025) | Doodling | Image → Image with Doodles | Ground Truth |
| Cat (Crawford, 2019) | Sticker Addition | Image → Image with Stickers | Human-labeled |
| PaintBucket (Dai et al., 2024) | Line Art Colorization | Line Art → Colored Image | Ground Truth |
| ReNé (Toschi et al., 2023) | Object Relighting | Image under Light A → Light A' | Ground Truth |

## B    ADDITIONAL DETAILS ABOUT TEST-TIME SCALING

### B.1    ATTENTION–GUIDED SEED SELECTION PROCEDURE

Algorithm 1 details the test-time seed selection procedure.

---

**Algorithm 1** Attention–Guided Seed Selection (AGSS)

---

**Require:** quad-grid $I$ with BR placeholder X, seeds $\mathcal{S}$, warmup steps $\mathcal{I}_{\text{probe}}=\{0,\ldots,i_{\text{warm}}\}$, critical blocks $\mathcal{B}^\dagger$

1: $c_{\text{vp}} \leftarrow \mathcal{E}([A, A', B])$
2: **for all** $s \in \mathcal{S}$ **do**                                      ▷ batched warmup
3:     $x_0^{(s)} \sim \mathcal{N}(0, I)$
4:     **for** $i \in \mathcal{I}_{\text{probe}}$ **do**
5:         advance one solver step on BR only; record $W_{s,i}^{(b,h)}$ for $b \in \mathcal{B}^\dagger$
6:     **end for**
7:     compute $\{p^{\text{br}}_{s,b,i}, p^{\text{vp}}_{s,b,i}\}$, then $D_{\text{br}}(s), D_{\text{vp}}(s)$
8: **end for**
9: $S_{\text{pivot}}(s) \leftarrow z(D_{\text{br}}(s)) - z(D_{\text{vp}}(s))$     (for all $s$, per-image $z$-norm)
10: $s^\star \leftarrow \arg\max_{s \in \mathcal{S}} S_{\text{pivot}}(s)$
11: continue denoising from the cached BR state of $s^\star$ to obtain $\hat{x}_0$; output $B' = \mathcal{D}(\hat{x}_0)$

---

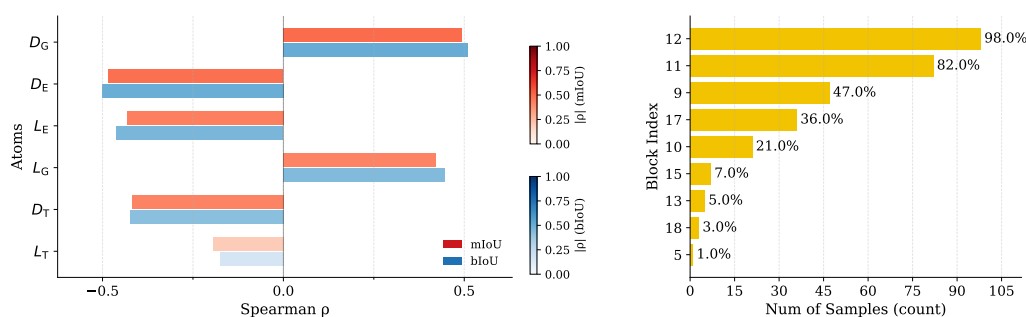

Figure 10: Spearman $\rho$ between $z$-normalized attention atoms and mIoU/bIoU across seeds.

Figure 11: Top-10 most seed-sensitive cross-attention blocks in Labs (2024).

## B.2 STATISTICAL ANALYSIS

**Notation.** Let $s \in \mathcal{S}$ be a seed, $b$ a cross-attention block, $h \in \{1, \ldots, H\}$ a head, and $i$ an early step in a warmup window $\mathcal{I}_{\text{probe}} = \{0, \ldots, i_{\text{warm}}\}$. Denote the attention by $W_{s,i}^{(b,h)} \in \mathbb{R}^{Q \times K}$. Queries are BR (target) tokens $\mathcal{Q}_{\text{br}}$; keys are partitioned into disjoint sets: $\mathcal{K}_{\text{br}}$ (BR), $\mathcal{K}_{\text{vp}}$ (context {A,A',B}), and $\mathcal{K}_{\text{txt}}$ (text). Average masses from BR queries at step $i$ are

$$p^\star_{s,b,i} = \frac{1}{H\,|\mathcal{Q}_{\text{br}}|} \sum_{h=1}^{H} \sum_{q \in \mathcal{Q}_{\text{br}}} \sum_{k \in \mathcal{K}_\star} W_{s,i}^{(b,h)}[q, k], \qquad \star \in \{\text{br, vp, txt}\}. \tag{7}$$

**Setting.** We evaluate 100 images $\times$16 seeds with BR-only warmup on steps $\{0, 1, 2\}$. For each (image, seed) we compute atoms over $\mathcal{B}^\dagger$ and $z$-normalize across the 16 seeds (per image). Performance is mIoU on the BR quadrant (bIoU yields the same ordering).

With $i^*=2$,

$$L_{\text{br}}(s) = \tfrac{1}{|\mathcal{B}^\dagger|} \sum_{b \in \mathcal{B}^\dagger} p^{\text{br}}_{s,b,i^*}, \qquad D_{\text{br}}(s) = \tfrac{1}{|\mathcal{B}^\dagger|} \sum_{b \in \mathcal{B}^\dagger} \left(p^{\text{br}}_{s,b,i^*} - p^{\text{br}}_{s,b,0}\right), \tag{8}$$

$$L_{\text{vp}}(s) = \tfrac{1}{|\mathcal{B}^\dagger|} \sum_{b \in \mathcal{B}^\dagger} p^{\text{vp}}_{s,b,i^*}, \qquad D_{\text{vp}}(s) = \tfrac{1}{|\mathcal{B}^\dagger|} \sum_{b \in \mathcal{B}^\dagger} \left(p^{\text{vp}}_{s,b,i^*} - p^{\text{vp}}_{s,b,0}\right), \tag{9}$$

$$L_{\text{txt}}(s) = \tfrac{1}{|\mathcal{B}^\dagger|} \sum_{b \in \mathcal{B}^\dagger} p^{\text{txt}}_{s,b,i^*}, \qquad D_{\text{txt}}(s) = \tfrac{1}{|\mathcal{B}^\dagger|} \sum_{b \in \mathcal{B}^\dagger} \left(p^{\text{txt}}_{s,b,i^*} - p^{\text{txt}}_{s,b,0}\right). \tag{10}$$

$L_\bullet$ capture *where* attention lands by the end of warmup; $D_\bullet$ capture *how* it pivots.

**Correlation map.** We compute Spearman rank correlations between $z$-normalized atoms and mIoU across seeds (pooled over images). As shown in Figure 10, the results align with our policy: BR (target) atoms are positively correlated with mIoU, visual-context atoms (from {A,A',B}) are negatively correlated, and text atoms exhibit weak correlations.

### B.2.1 IDENTIFYING CRITICAL BLOCKS

Seed sensitivity concentrates in a narrow stage. For each image and block $b$, we define a simple visual context–target gap

$$\text{gap}_{b,i}^{(s)} = p_{s,b,i}^{\text{vp}} - p_{s,b,i}^{\text{br}}, \quad i \in \{0, 1, 2\}. \tag{11}$$

We scan cross-attention blocks and summarize each block by two statistics over the warmup: (i) a level term (the average context–target gap), and (ii) a growth term (the change in the gap from the first to the last warmup step). Blocks whose level/growth exhibit high variance across seeds are most discriminative for seed ranking. We therefore retain the top few blocks as the critical set $\mathcal{B}^\dagger$ (we use $K{=}3$). Aggregated across images, the same small stage (e.g., blocks $\{9, 11, 12\}$) consistently emerges (Figure 11).

## C ADDITIONAL STUDIES

### C.1 EFFECT OF VISUAL PROMPTS

In Section 4.2, we demonstrate that PICO supports free-form inputs for personalized segmentation tasks. Here, we further investigate the role of visual prompts—*i.e.*, in-context input-output exemplars ($A \rightarrow A'$)—in providing fine-grained control over output behavior across additional representative task categories. Given a fixed query image $B$ and text prompt, PICO flexibly adapts to different task intents by interpreting the visual demonstration ($A \rightarrow A'$), producing diverse and context-appropriate outputs $B'$.

**Context-Aware Sticker Addition.** Figure 12 shows how the visual prompt controls what customized object or doodling is added, where it is placed, and how it is scaled. For example, the size of a Christmas hat (Row 2) changes based on the visual exemplar, despite the same text prompt ("Add the sticker"). This task highlights the limitations of text-only instructions and the strength of visual exemplars for conveying spatial and compositional intent.

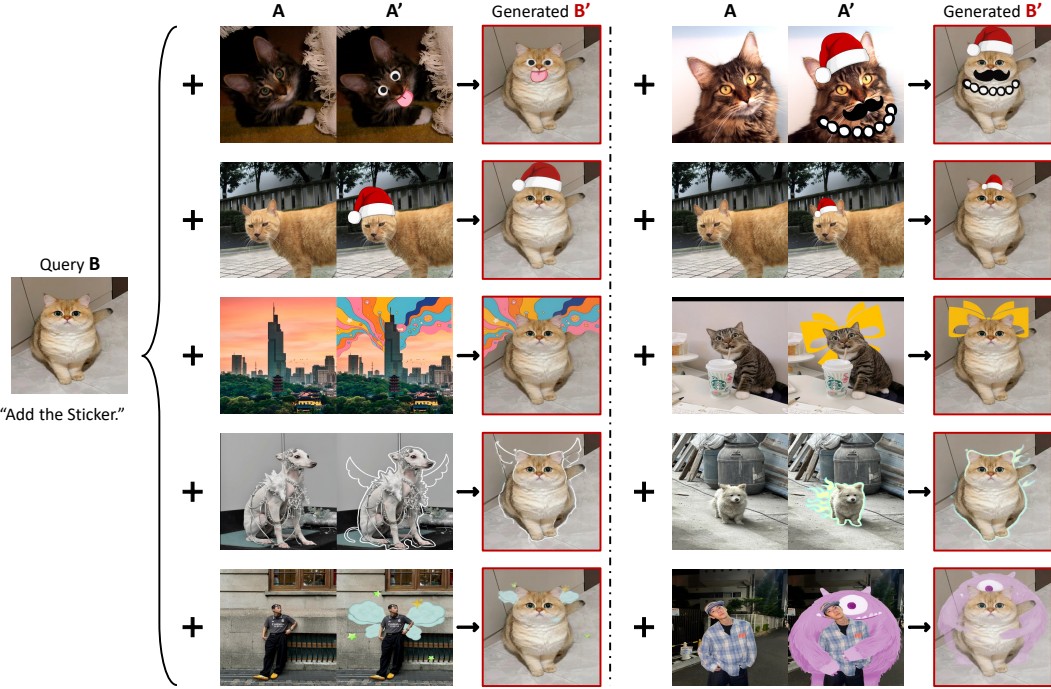

Figure 12: **Context-aware sticker addition with visual prompt control.** Given the same query image $B$ and text prompt ("Add the sticker"), PICO generates diverse outputs $B'$ solely based on visual prompt ($A \rightarrow A'$). The model captures variation in object type, position, and scale, demonstrating precise spatial and semantic interpretation from visual prompts.

**Personalized Edge Detection.** As shown in Figure 13, PICO handles edge detection tasks defined by spatial constraints and style cues in the visual prompts. The model is able to adaptively predict edges of specific regions (*e.g.*, top vs. bottom) or emulate particular edge styles (*e.g.*, Canny vs. Euclidean vs. texture-based) without making any changes to the text prompt ("Predict the edges").

These results confirm that PICO effectively comprehends the visual relation conveyed by the in-context input-output pairs, and applies the underlying visual logic to query images. The quad-grid in-context format provides a strong structural prior for visual reasoning, enabling flexible, controllable, and free-form at test time.

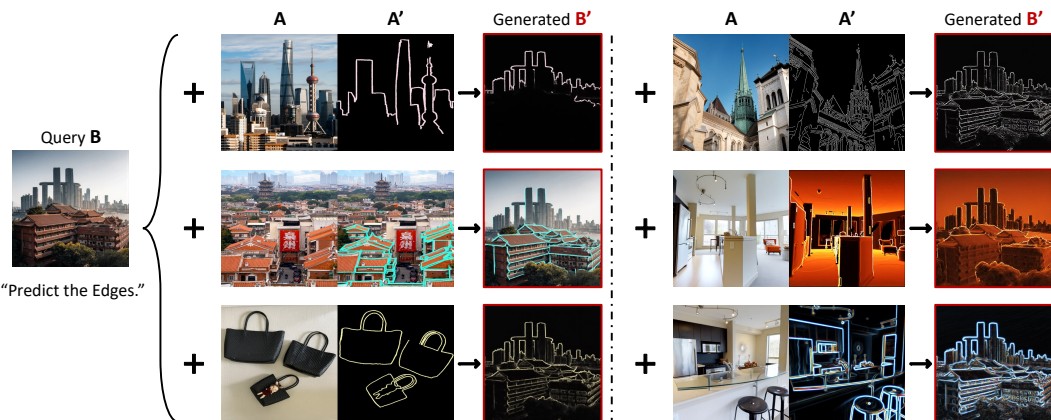

Figure 13: **Personalized edge detection with visual prompt control.** Given the same query image $B$ and text prompt ("Predict the edges"), PICO generates diverse outputs $B'$ solely based on visual exemplars $(A \rightarrow A')$. The model adapts spatial focus (*e.g.*, top or bottom) and edge style (*e.g.*, canny, euclidean, texture), guided entirely by visual cues.

Table 6: **Ablation on task-type effects for personalized segmentation.** The best results are highlighted in **bold**, and the second-best are underlined.

| Method | PerSeg | | | DOGS | | | PODS | | | PerMIS | | |
|--------|mIoU↑|bIoU↑|F1↑|mIoU↑|bIoU↑|F1↑|mIoU↑|bIoU↑|F1↑|mIoU↑|bIoU↑|F1↑|
| PICO | 90.97 | 76.13 | 62.82 | 71.02 | 54.71 | 49.84 | **68.72** | **60.26** | 44.88 | 49.52 | **33.63** | 14.90 |
| Remove physical | 82.68↓ | 72.45↓ | 54.11↓ | 62.47↓ | 52.56↓ | 51.36↑ | 50.44↓ | 50.97↓ | 35.96↓ | 42.12↓ | 27.72↓ | 12.02↓ |
| Remove low-level | 89.71↓ | 76.14↑ | 62.89↑ | 75.52↑ | 60.86↑ | 60.47↑ | 61.29↓ | 59.52↓ | **45.91**↑ | 48.32↓ | 32.60↓ | 15.22↑ |
| Remove Generative | 70.88↓ | 59.68↓ | 39.15↓ | 63.02↓ | 52.11↓ | 44.26↓ | 22.86↓ | 23.29↓ | 11.38↓ | 35.07↓ | 21.56↓ | 7.37↓ |
| Remove Semantic | **92.90**↑ | **78.11**↑ | **63.98**↑ | 74.93↑ | 58.54↑ | **61.19**↑ | 62.07↓ | 57.64↓ | 44.41↓ | **50.58**↑ | 32.58↓ | **16.53**↑ |

## C.2 TASK TYPE ABLATION

Our VisRel dataset balances diversity across task types to holistically support personalization. We assess each family's contribution by removing three tasks per family (30 samples) and retraining under identical settings. Results in Table 6 show:

- **Physical/geometric tasks** (*e.g.*, depth, reshading, 3D keypoints) consistently help. Removal reduces performance. 3D/spatial reasoning improves object boundary awareness.
- **Generative tasks** (*e.g.*, doodling, relighting, line-art colorization) are critical. Removal causes big performance drop. These high-semantic/local tasks teach object-aligned editing essential for segmenting user-specific objects.
- **Low-level tasks** (*e.g.*, deblurring, dehazing, low-light enhancement) are neutral but still valuable. Removal shows small changes. While not directly beneficial, they don't harm performance, validating our inclusive design.
- **Semantic perception tasks** (*e.g.*, stuff segmentation, object detection, affordance) can be conflicting. Interestingly, removal improves results. We hypothesize that class-level labels may suppress fine instance-level distinctions, which are essential for personalized segmentation.

## C.3 Spatial Position of the Grid Format

During training and inference, we fixed the placeholder position (i.e., the cell of $B'$) in the $2 \times 2$ grid. To evaluate the impact of placeholder positioning, we conducted the experiment by changing the curernt horizontal layout (*Top–Bottom*, TB) to a vertical arrangement (*Left–Right*, LR), using identical datasets.

The results in Tables 7, 8 show that grid positioning has minimal impact on overall model performance. In some cases, the LR layout improves personalized segmentation metrics. The grid layout (or, essentially, the positional embedding) has little impact, but the learned visual-relation space drives performance.

Table 7: **Ablation on grid layout for personalized image segmentation.**

| Method | PerSeg | | | DOGS | | | PODS | | | PerMIS | | |
|---|---|---|---|---|---|---|---|---|---|---|---|---|
| | mIoU↑ | bIoU↑ | F1↑ | mIoU↑ | bIoU↑ | F1↑ | mIoU↑ | bIoU↑ | F1↑ | mIoU↑ | bIoU↑ | F1↑ |
| PICO (TB) | 90.97 | **76.13** | 62.82 | 71.02 | 54.71 | 49.84 | 68.72 | 60.26 | 44.88 | **49.52** | 33.63 | 14.90 |
| PICO (LR) | **91.73** | 75.87 | **63.53** | **75.90** | **58.44** | **58.71** | **70.27** | **61.79** | **45.88** | 47.69 | **33.64** | **17.34** |

Table 8: **Ablation on grid layout for personalized test-time task generalization.**

| Method | (a) *Deraining with Inpainting* | | (b) *Inpainting with Stylization* | | | |
|---|---|---|---|---|---|---|
| | PSNR↑ | SSIM↑ | Gram↓ | FID↓ | LPIPS↓ | ArtFID↓ |
| Ref | $\infty$ | 1.00 | 17.29 | 1.71 | 0.62 | 4.38 |
| PICO (TB) | 22.24 | **0.67** | **21.27** | 1.87 | 0.52 | **4.38** |
| PICO (LR) | **22.42** | **0.67** | 21.51 | 1.87 | **0.51** | 4.39 |

# D More on Test-time Tasks Generalization

## D.1 Quantitative Evaluation

To complement the main results in Section 4.3, we provide full details of the quantitative setups. We evaluate two representative composite tasks:

**(1) Deraining with inpainting.** We evaluate 200 images corrupted by rain and occlusions. We use: (i) PSNR to assess pixel-level reconstruction fidelity, and (ii) SSIM to measure structural similarity between the predicted output $B'$ and the clean reference image Cleaned($B$).

**(2) Inpainting with Stylization.** We evaluate 265 stylized images across 40 style different styles, each stylized using StyleID (Chung et al., 2024) and then corrupted by watermarks or inpainting masks. Evaluation metrics include: (i) Gram Matrix Distance between $B'$ and the reference style image $A'$ to measure style fidelity, (ii) LPIPS between $B'$ and the original Cleaned($B$), to evaluate content preservation and occlusion removal, and (iii) ArtFID (Chung et al., 2024), defined as (LPIPS+1)·(FID+1), which captures the overall trade-off between perceptual faithfulness and style fidelity. As a reference upper bound, we include the "ground truth" result: applying StyleID (Chung et al., 2024) directly to the clean image Cleaned($B$) using the same target style as $A'$.

## D.2 Additional Qualitative comparisons

Figures 14, 15, 16, 17, 18 present additional qualitative comparisons across diverse test-time personalized tasks: background-only stylization, edge detection with spatial constraints, joint deraining with inpainting, watermark removal with stylization, and context-aware sticker addition. PICO demonstrates consistent superiority in aligning with the task intent, as defined by in-context visual exemplar pair ($A \to A'$). GPT-4o shows strong semantic-level understanding but lacks precision in content fidelity and spatial alignment, especially in tasks that require geometric fidelity or pixel-aligned outputs.

## E    ADDITIONAL RESULTS

We present additional results generated by PICO across diverse tasks in Figure 19. For personalized
face parsing (Figure 19(c)), PICO leverages contextual appearance cues to consistently segment se-
mantically identical components. Despite never being trained on facial data, the model performs
well on this out-of-domain setting, demonstrating robustness and flexibility. PICO also supports a
broad range of standard visual tasks spanning restoration, perception, and generation, as illustrated
in Figure 19(d–k). While trained on these tasks, PICO generalizes effectively to novel object in-
stances and scenarios with as few as 10 example pairs per task. Notably, for object relighting, *i.e.*,
transforming an object under one lighting condition into another, PICO predicts physically plausi-
ble shadows aligned with previously unseen query objects (Figure 19(f)). These results suggest an
implicit understanding of lighting and object interactions.

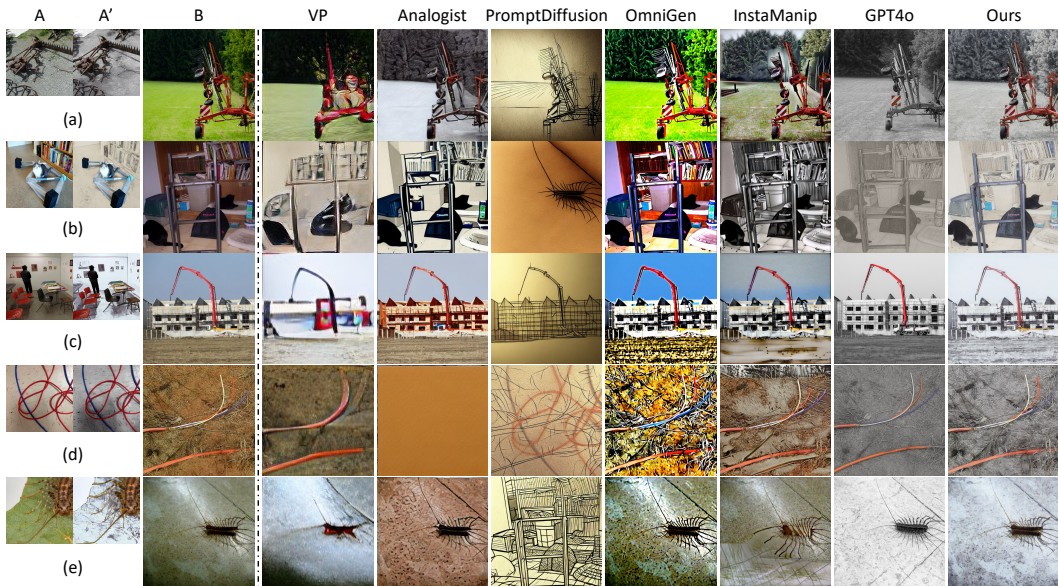

Figure 14: **Qualitative comparisons on background-only stylization.** PICO selectively stylizes
the background while preserving the foreground.

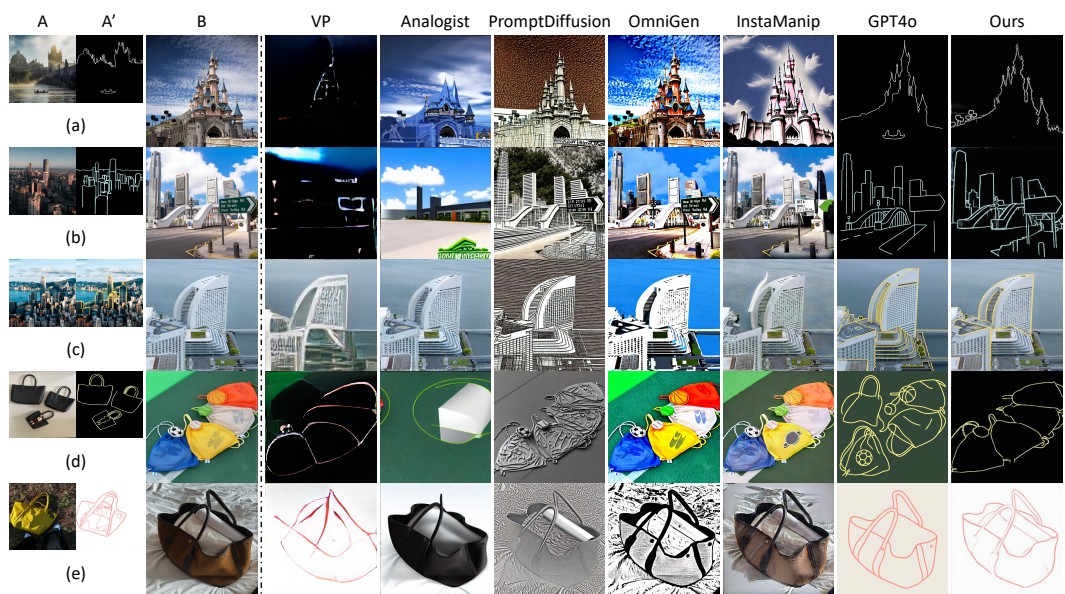

Figure 15: **Qualitative comparisons on edge detection with spatial constraints.** PICO accurately predicts personalized edge maps guided by the visual prompt.

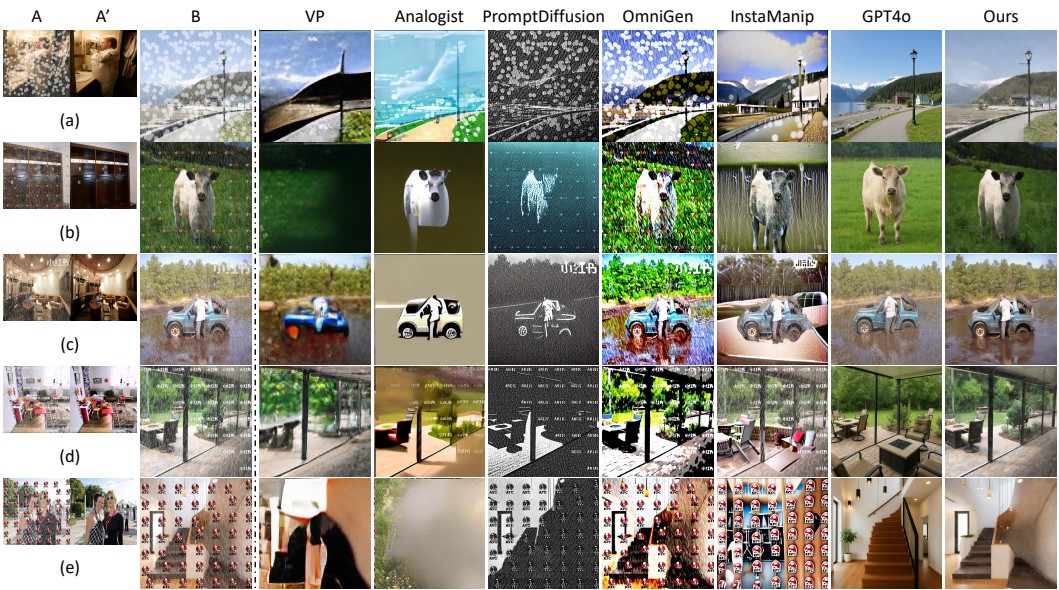

Figure 16: **Qualitative comparisons on joint deraining with inpainting.** PICO removes both rain and occlusions simultaneously.

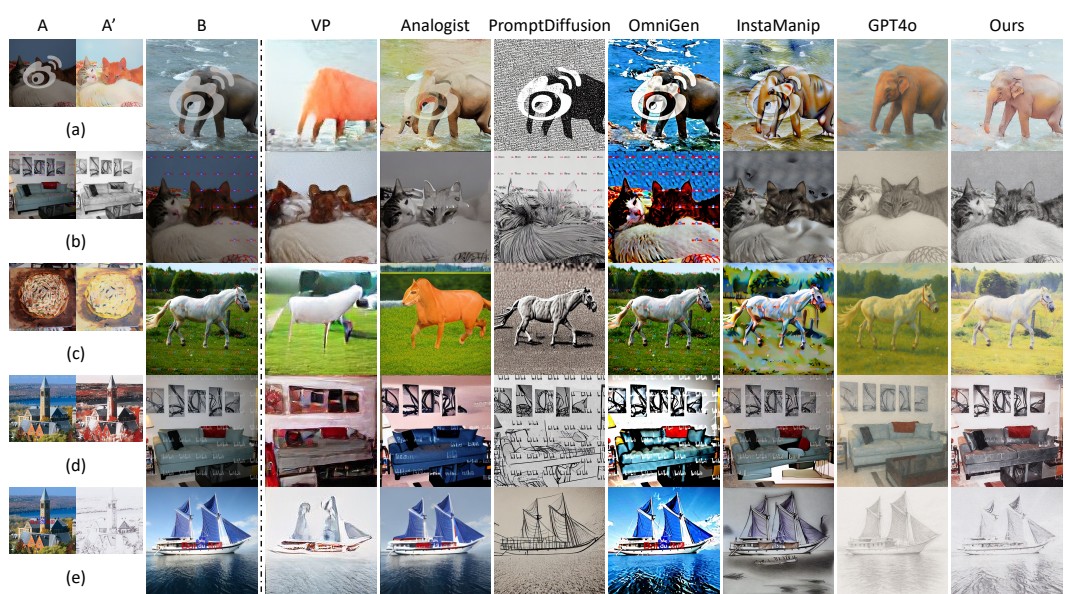

Figure 17: **Qualitative comparisons on watermark removal with stylization.** PICO removes occlusions while transferring target style.

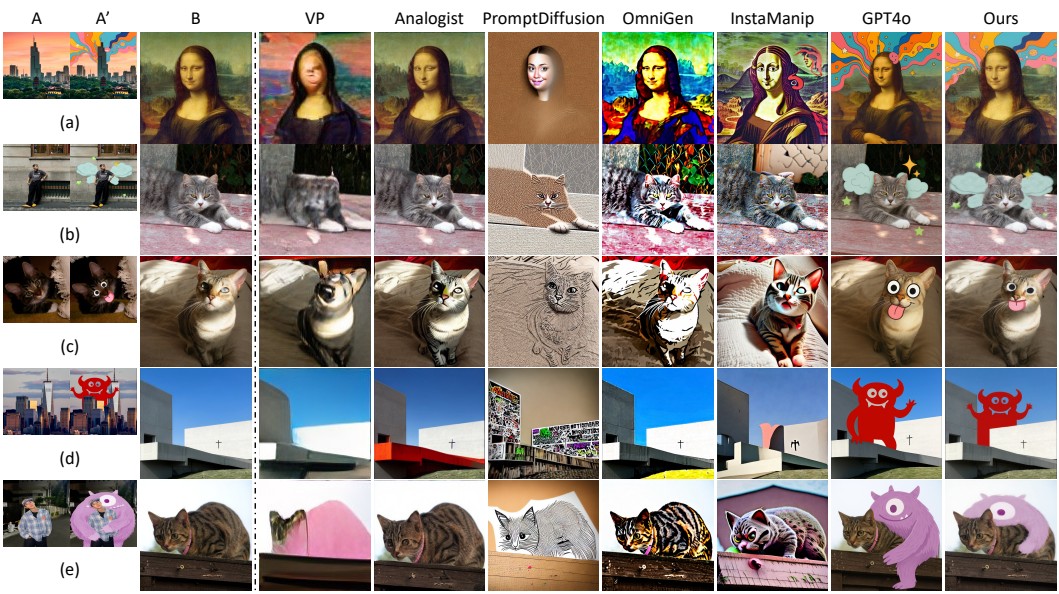

Figure 18: **Qualitative comparisons on context-aware sticker addition.** PICO learns from the visual exemplar where and how to place the sticker (*e.g.*, object type, size, position).

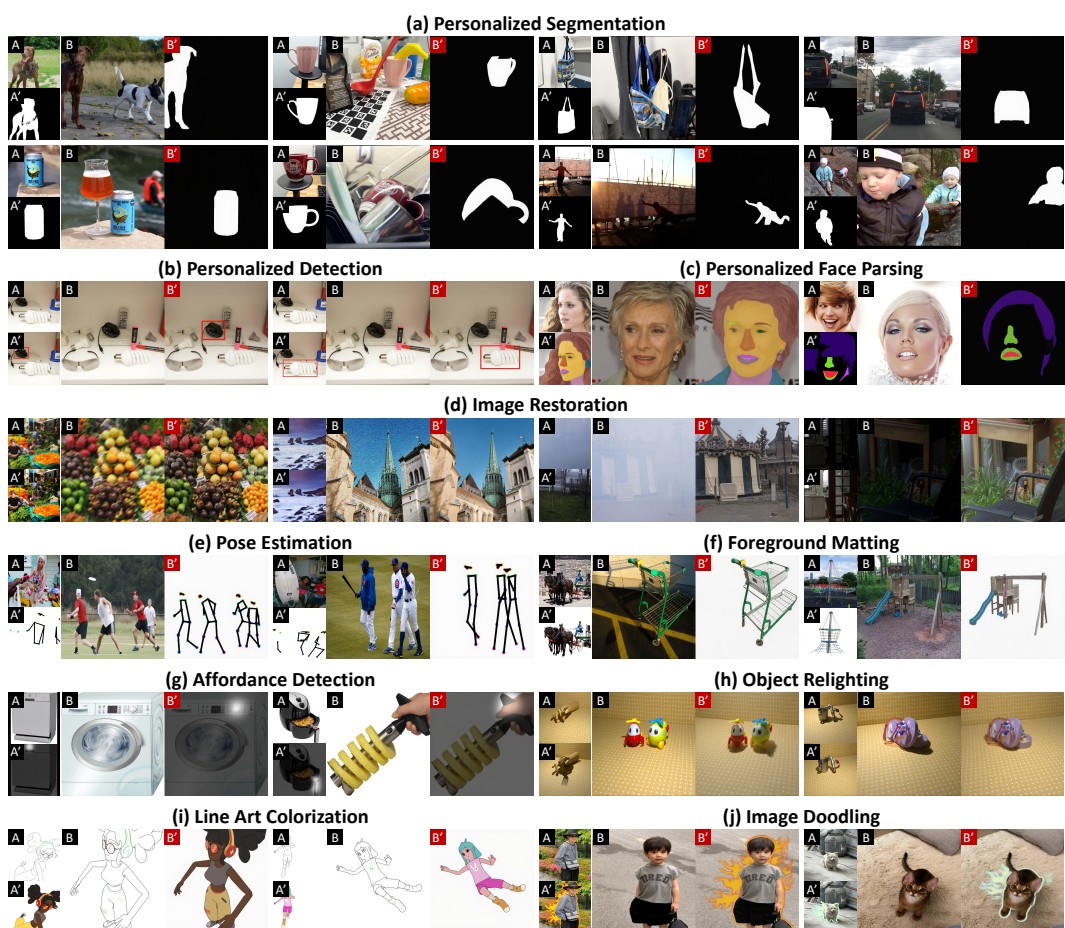

Figure 19: **Additional results generated by PICO.**

