# OpenReview forum: "Personalized Vision via Visual In-Context Learning"
_ICLR.cc/2026/Conference — Submitted to ICLR 2026_

### Official Review · Reviewer_Cvqg · 2025-10-30

**Soundness:** 2
**Presentation:** 3
**Contribution:** 2
**Rating:** 4
**Confidence:** 3

**Summary:**

This paper proposes PICO, a unified framework for visual in-context learning using a four-panel image format and a diffusion-based generative backbone.

**Strengths:**

- The authors propose a new dataset that consolidates a diverse set of vision tasks into a unified training corpus. This supports the model’s ability to generalise to various task formats under a single image-to-image interface. While the dataset is constructed from existing sources, it is well-structured and contributes to the feasibility of training a universal visual prompting model.

- The model performs competitively across several custom personalisation tasks and evaluates against a broad range of recent baselines.

**Weaknesses:**

- A good dataset is not merely a collection of existing datasets. It must demonstrate scientific value through structure, diversity, and measurable challenge. While the dataset proposed in PICO is well-organised and supports a wide range of visual tasks, it lacks rigorous analysis of task balance, generalisation limits, or empirical ablations on the selected tasks or images. As such, I do not consider the dataset contribution to be of standalone scientific value.

$~$

- The proposed framework represents an incremental step rather than a fundamentally novel invention. The four-panel in-context format and image-to-image task unification have already been explored in prior works, such as Analogist (Gu et al., 2024, ACM TOG). Similarly, the concept of attention-guided seed selection has appeared in several earlier works and is not introduced here as a fundamentally new technique.

$~$

- The paper's comparisons conflate off-the-shelf models with PICO, which is trained and fine-tuned on a custom dataset. This makes the evaluation setup less fair and diminishes the strength of the comparative claims.

$~$

- The authors argue that their quad-grid format addresses visual task heterogeneity, I do not agree with the strength of this claim (lines 165–168). However, this approach renders all outputs as images which is an effective workaround, but not a principled solution to the problem of heterogeneous output types. It avoids cases that cannot be expressed visually, such as visual question answering or counting (e.g., "how many cats are in the image?"), which the current design cannot accommodate.

$~$

- Although certain benchmarks are held out during training, the underlying task types, segmentation, detection, inpainting, are already present in the training data. Therefore, the evaluations measure generalisation to new instances, not to new task types, weakening the paper's broader generalisation claims.

$~$

- The paper uses a very strong diffusion-based backbone. While this helps showcase what the framework can achieve, without controlling for model size or architectural advantage, it becomes difficult to isolate the impact of the proposed method (and dataset).

$~$

- [Minor] There are several grammar and formatting issues: e.g., Extra “and” in line 209; Two commas in line 211; In general, the writing needs polishing.

$~$

My final remarks here would be that this paper is ambitious and presents a well-engineered system for visual in-context learning. However, its core contributions are built on prior work, with no clear methodological innovation. The performance gains are difficult to attribute given the use of a strong diffusion model and further finetune it, unlike many baselines that rely on off-the-shelf models without even finetuning. Without a disentanglement of these factors, and given that the dataset is primarily a collection of existing sources, I do not find the contribution sufficient for publication.

**Questions:**

1- How does your framework offer technical novelty beyond prior works?

$~$

2- Given that your model is finetuned on a custom dataset while baselines use off-the-shelf backbones or trained on specific tasks, how do you ensure a fair evaluation?

$~$

3- Can you provide ablations or evidence to disentangle the performance gains from your backbone scale versus the proposed in-context learning mechanism?

$~$

4- What evidence supports the dataset's scientific contribution beyond being a curated aggregation of existing datasets?

$~$

5- If your evaluation benchmarks involve **task types** already present in training (e.g., segmentation), how do you justify claims of generalisation to novel tasks?

$~$

I hope that clear answers to these questions will help clarify the paper's core contributions and allow for a more informed and fair assessment of its scientific merit.

**Details Of Ethics Concerns:**

The authors propose a dataset composed entirely of existing open-source benchmarks.

---

> ### Author Response · Authors · 2025-11-23
> **(1/2)**
>
> Thank you for your detailed review and for recognizing the well-engineered system, the structured VisRel dataset, and the competitive, broad performance of our method. We address the questions in detail below; please let us know if there are further comments or concerns.
>
> > **What is actually new? and our contribution**
>
> Our novelty is at the **problem + system** level: how we formulate **personalized vision** as visual ICL, and how we turn a generative prior into a one-shot personalized operator.
>
> 1. Prior viusal ICL work does not address test-time generalization.
>     - Existing Visual ICL methods [1, 2] focus on solving predefined tasks seen during large-scale pretraining. They do **not** evaluate generalization to new tasks or new objects at test time, which is the core goal of ICL.
>     - Analogist [3], specifically, is training-free and therefore restricted to Stable Diffusion’s RGB output space. It cannot handle dense or discriminative prediction tasks, whose outputs lie in other spaces such as depth maps, normal maps or segmentation masks. while PICO's lightweight finetuning to repurpose the generative prior, unlocking dense/discriminative tasks (e.g., depth/normal estimation, pose estimation), while also supporting the RGB-space generation tasks.
>
> 2. We target the personalized vision setting for the first time (objects and tasks).
>
>     - We study the challenge of adapting at test-time, from a single exemplar, to user-defined objects and tasks under severe data constraints. To our knowledge, this is a scenario unaddressed by previous methods.
>     - Prior “personalized vision” work focuses on personalized objects within a single task (e.g., instance segmentation) [4, 5, 6], and methods such as PRPG [4] use generative priors to synthesize more object-specific data for training personalized representations, which is very computational expensive. PICO instead treats the generative model iteself as a visual in-context learner, allowing flexible test-time adaptation.
>
> 3. Our solution is tailored to this setting and yields new insights.
>
>     - Within this formulation, we show that **task diversity matters more than sheer data volume** for visual ICL, and we design our VisRel dataset and ablations around this finding.
>     - Technically, we also introduce an **attention-guided seed scorer specifically for this quad-grid personalized setting**, to mitigate the stochastic failure modes of generative priors on deterministic tasks.   To our knowledge, this is the first use of cross-attention over a visual ICL grid for seed selection, and the mechanism is orthogonal and reusable, e.g.,  for other visual ICL/editing frameworks and for choosing effective visual prompts.
>
> Overall, our contribution is a system-level novelty: a new problem setting, a practical solution, and a structured evaluation that lets us analyze how task diversity affects personalized in-context learning.
>
> > **Fairness of comparisons and evaluation setup**
>
> We agree this is a valid concern. Since our work targets a new setting, i.e, evaluating the ability to solve unseen personalized tasks (both object- and task-level), there is no perfectly aligned prior baseline, so we tried our best to ensure fairness by selecting the strongest available baselines and covering a broad range of methods.
>
> For example, in Personalized image segmenation: we include large-scale pretrained models, the prior personalized segmenation approaches , and visual in-context methods. To make the setup transparent, Table 2 explicitly annotates, for each method, the training data it uses, its training objective, and its backbone.
>
> Finally, our ablation studies to isolate contributions within our method, and to attribute performance gain to our method.
>
>
> > **Backbone strength vs ICL mechanism**
>
> We agree that the FLUX with a strong generative prior and DiT architecture provides a powerful starting point. Our ablations, however, are conducted within the backbone to isolate what actually drives the gains:
> (1) Figure. 6 shows that **task diversity** is the key for robust generalization to novel, user-defined tasks, more so than sheer data volume;
> (2) Table 4, and Figure 5 show that **minimal text prompts** serve as soft task boundaries and are important for stable behavior under mixed relations;
> (3) Table. 1 and Figure. 7 quantify and visualize how **our attention-guided seed scorer** improves deterministic tasks by mitigating randomness from the generative prior;
> (4) Table. 6 studies **task-type effects** for personalized segmentation; and
> (5) we also test **different grid layouts** in Table. 7 and 8.
>
> Taken together, these controlled ablations indicate that the gains come from our ICL formulation (VisRel + prompts + seed scorer) and how we repurpose the generative prior, rather than from backbone scale alone, and we expect these insights to transfer to other backbones.

---

> > ### Author Response · Authors · 2025-11-23
> > **(2/2)**
> >
> > > **Scientific value of the VisRel dataset**
> >
> > The contribution is not raw collection, but **how** the data is selected and formatted for personalized visual ICL. The key design principle is inter- and intra-task diversity in a shared “visual-relation” space, which we find crucial for test-time generalization.
> >
> > For example, if we train only on standard inpainting with fixed black square masks, the model fails to generalize well to real-world watermark removal tasks with arbitrary shapes and transparencies. In contrast, VisRel's carefully curated task combinations are balanced, with minimal text prompts as soft task boundaries, enabling the model to perform these behaviors with only a few examples and encouraging it to learn general **relation** patterns.
> >
> > We emphasize VisRel's role as a **principled template**: given PICO’s data-efficient training, users can construct their own “visual-relation spaces” tailored to new domains by following the same diversity-driven design.
> >
> > > **Generalization to “novel tasks”**
> >
> > Our claims about generalization are made in two personalized vision setting:
> > - new objects/instances:
> >
> >     Personalized segmentation benchmarks (PerSeg, PODS, PerMIS, DOGS) contain objects that never appear in VisRel. Here, PICO adapt to new instance from a single exemplar, without any per-subject finetuning.
> > - novel tasks specifications at test time:
> >
> >     **In Sec. 4.3**, we evaluate test-time personalization on user-defined visual tasks that differ from conventional CV setups and are **not** introduced as separate tasks during training, including composite tasks, spatially constrained tasks, and semantic-conditional tasks. PICO solves these by infering the user intent from its demonstration at test time, again without retraining.
> >
> > > **Quad-grid format and task heterogeneity**
> >
> > We agree that the current quad-grid design is **not** a principled solution to fully heterogeneous output types, especially those that are inherently verbal. Our claim is more modest: within the scope of **image-to-image tasks**, the quad-grid format provides a practical way to unify very different **visual** outputs (e.g., segmentation masks, depth maps, stylized images, edited images) under a single operator and training objective. We do **not** intend to claim coverage of tasks whose outputs are inherently non-visual, such as visual question answering, counting (“how many cats are in the image?”), or free-form text.
> >
> > (Extending this in-context formulation beyond image-to-image to VQA or counting, inspired by recent advances [7] where models can "draw" the mathematical solution visually, is an interesting direction for future work, but is beyond the scope of this paper.)
> >
> >
> > > **Writing and minor issues**
> >
> > We have corrected the formatting issues (the extra "and" and double comma) in the revised revision. Thank you for your attention to detail!
> >
> >
> > **References:**
> >
> > [1] Bar, Amir, et al. "Visual prompting via image inpainting." NeurIPS, 2022.
> >
> > [2] Yang, Yifan, et al. "Imagebrush: Learning visual in-context instructions for exemplar-based image manipulation." NeurIPS, 2023.
> >
> > [3] Gu, Zheng, et al. "Analogist: Out-of-the-box visual in-context learning with image diffusion model." ACM TOG, 2024.
> >
> > [4] Sundaram, Shobhita, et al. "Personalized Representation from Personalized Generation." ICLR 2024.
> >
> > [5] Nguyen, Thao, et al. "Yo'Chameleon: Personalized Vision and Language Generation." CVPR, 2025.
> >
> > [6] Samuel, Dvir, et al. "Where's Waldo: Diffusion Features For Personalized Segmentation and Retrieval." NeurIPS, 2024.
> >
> > [7] Google DeepMind. (2025). Gemini 2.5 Flash Image (Nano Banana). https://deepmind.google/models/gemini-image/flash/

---

### Official Review · Reviewer_H62u · 2025-10-31

**Soundness:** 3
**Presentation:** 2
**Contribution:** 2
**Rating:** 4
**Confidence:** 2

**Summary:**

This paper presents PICO (Personalized In-Context Operator), a framework for personalized vision. The authors reformulate personalization as a visual in-context learning (ICL) problem. In this setting, the model can understand and perform user-defined visual tasks during testing without any extra training or fine-tuning. The method uses a four-panel input format. An example image pair, written as (A → A′), defines the task, and a query image B is given for applying the same transformation, which produces B′. This allows the model to learn the transformation from only one example. To train the system, the authors design a VisRel dataset, which contains a small but diverse set of 27 visual tasks such as image restoration, segmentation, depth estimation, and style transfer. They also use an attention-guided seed scorer that improves the stability and reliability of generative inference by choosing the most effective random seeds.
It can generalize to new user-defined tasks.
It achieves strong results in both recognition and generation tasks with very limited training data.
The VisRel dataset, showing that task diversity is more important than dataset size for strong generalization.
The attention-guided seed scorer, which improves inference consistency without any retraining.

**Strengths:**

The paper is generally clear and organized, with a logical flow between sections. Figures and diagrams are well designed and help explain the method. However, there are a few small writing and formatting issues in the text, such as spacing and repeated citations, which should be corrected for better readability.
The experiments are of good quality and cover a range of visual tasks, showing that the model performs consistently across segmentation, generation, and editing. The use of attention-guided seed scorer improves inference stability, even if it is a heuristic approach. The focus is on task variety rather than size, which maybe a good direction for visual in-context learning research.

**Weaknesses:**

While the paper outlines a diffusion-based visual in-context learning framework, the data flow between components (VAE encoder, diffusion transformer, and LoRA modules) is not clearly explained. The description of the “quad-grid” input format could be supported with more explicit architectural diagrams or pseudocode.
The paper’s main idea of using exemplar pairs (A, A′) and a query image (B) to predict a target (B′) has already been explored in other works, such as "Analogist: Out-of-the-box Visual In-Context Learning with Image Diffusion Model" and "ImageBrush: Learning Visual In-Context Instructions for Exemplar-Based Image Manipulation". Similar to Analogist, PICO also use inference-based visual ICL with light text cues and transfer attention relations through diffusion models. Because of these strong similarities, the idea in this paper does not appear to be very novel.
There are minor writing issues, such as no space between vs. and colorful in line 185, and a duplicate citation in lines 113–114 written as Bar et al. (Bar et al., 2022).


Although the experiments are detailed, the comparison could be clearer if the authors also included results for PRPG. The paper mentions that generating per-instance synthetic data is costly and hard to scale, but adding even limited results for these cases would make the evaluation more complete.
The GPT-4o results are based on only ten random samples due to API limits. Using more samples or a consistent setup would improve fairness and reliability.

**Questions:**

While PICO builds on similar ideas to Analogist and ImageBrush, which also leverage exemplar pairs and attention transfer for diffusion-based visual in-context learning, could the authors elaborate on the specific innovations introduced by PICO beyond adopting a diffusion transformer in place of the U-Net backbone?

The description of how the VAE encoder, diffusion transformer, and LoRA modules interact is unclear. Could the authors include a more detailed diagram or short pseudocode to show how {A, A′, B} are encoded and passed through the model?

Since PRPG results were omitted due to cost, could the authors share small or partial results to show scalability and fair comparison?

The GPT-4o evaluation used only ten random samples. How consistent are the results with different samples, and could more data improve reliability?

---

> ### Author Response · Authors · 2025-11-22
> **(1/2)**
>
> Thank you for your valuable review and for recognizing the clarity and organization of the paper, the quality and breadth of our experiments, the promise of focusing on task variety rather than sheer data size for visual ICL, and the usefulness of our attention-guided seed scorer in stabilizing inference. We address the questions in detail below; please let us know it there are further comments or concerns.
>
> > **Novelty against Analogist / ImageBrush”**
>
> While PICO shares the "example-based" interface with Analogist [1] and ImageBrush [2], its goals and mechanisms are quite different and more general:
>
> - Analogist is training-free and therefore restricted to Stable Diffusion’s RGB output space. It cannot handle dense or discriminative prediction tasks, whose outputs lie in other spaces such as depth maps, normal maps or segmentation masks. while PICO's lightweight finetuning to repurpose the generative prior, unlocking dense/discriminative tasks (e.g., depth/normal estimation, pose estimation), while also supporting the RGB-space generation tasks.
>
> - ImageBrush, similarly, is designed for RGB semantic editing and is trained on four “in-the-wild” video datasets and a large synthetic manipulation dataset, and primarily targets editing/manipulation within its training domains. PICO instead focuses on test-time generalization to novel user-defined tasks beyond training data.
>
> > **Data flow clarity: VAE, diffusion transformer, LoRA, quad-grid**
>
> Below is pseudocode showing how the VAE, diffusion transformer, LoRA modules, and the quad-grid interact.
> We will open-souce the code and pretrained model checkpoints upon publication.
>
> ```
> # ------- Train phase ------- #
> # Input Grid: [A  A′
> #              B  B′ ]
> grid = concat_2x2(A, A', B, B')
> z = ae.encode(grid) # VAE encode full grid → latent z
>
> # Flow matching noise
> z_noisy = (1 - t) * z + t * eps
>
> # Restore the three context quadrants from clean z
> z_noisy[TL] = z[TL]      # A
> z_noisy[TR] = z[TR]      # A'
> z_noisy[BL] = z[BL]      # B
> # BR kept noisy          # B'
>
> # DiT with LoRA adapters
> eps_hat = Flux_DiT_LoRA(z_noisy, text_tokens)
>
> # Loss is computed only on target quadrant B'
> loss = || eps_hat[BR] - eps[BR] ||²
>
> # ------- Infernce phase ------- #
> # Bottom-right cell X is a placeholder (black or any)
> grid = concat_2x2(A, A', B, X)
> z = ae.encode(grid)
> z_noisy = z.clone()
>
> # A, A', B stay clean; B' is intialized as Gaussian noise
> z_noisy[BR] = eps
>
> # Interatively denoise
> for t in timesteps:
>     eps_hat = Flux_DiT_LoRA(z_noisy, text_tokens, t)
>     z_noisy = update(z_noisy, eps_hat, t)
>
> B'_pred = ae.decode(z_noisy)[BR] # VAE decode
> ```
>
> > **Comparison with PRPG**
>
> Our current evalution of PRPG [3] is:
>
> - For DOGS and PODS (PRPG's own datasets), we **do** include PRPG results in Table 1. We use the synthetic DreamBooth-augmented and negative examples released by the authors to train the personalized representations, and then perform personalized segmentation. These results already show that PICO performs on par with or better than PRPG in the setting PRPG was designed for, **making this a fair and direct comparison**.
>
> - For other dataset, e.g., PerMIS [4], each instance has only one example. In this setting, training a reliable Dreambooth model per object (and preparing the corresponding negative data) is highly unstable and computationally expensive, so the PRPG pipeline becomes a practical bottleneck.
>
> - Instead, PICO performs personalization entirely at test time via in-context examples, without any per-instance retraining. The absence of PRPG results on PerMIS **reflects a practical limitation of this method**, and highlights **the need for a more flexible and efficient personalized vision pipeline**, which is what PICO addresses.
>
> > **GPT-4o evalutation**
>
> We agree that the GPT-4o experiment is small-scale and should be interpreted with caution. We report the mean ± standard deviation across the 10 random samples:
>
> | Task                         | Metric       | GPT-4o*            | PICO (Ours) |
> |------------------------------|--------------|--------------------|-------------|
> | *deraining with inpainting*| PSNR ↑ (dB)  | 12.29 ± 1.97       | **22.24**   |
> |                              | SSIM ↑       | 0.257 ± 0.153      | **0.67**    |
> | *inpainting with stylization* | Gram ↓   | 22.04 ± 3.76       | **21.27**   |
> |                              | FID ↓        | 1.87 ± 0.19        | **1.87**    |
> |                              | LPIPS ↓      | 0.682 ± 0.12       | **0.52**    |
> |                              | ArtFID ↓     | 4.81 ± 0.23        | **4.38**    |
>
> Even considering this variance, the GPT-4o scores remain lower than PICO’s (e.g., PSNR and ArtFID), so the qualitative gap between them is robust. We use GPT-4o mainly as a reference for task difficulty, while our key claims are grounded in full comparisons with open-source baselines. In the final version, we plan to increase the sample size.

---

> > ### Author Response · Authors · 2025-11-22
> > **(2/2)**
> >
> > > **Writing and minor issues**
> >
> > We have corrected the formatting issues (spacing around "vs.", duplicate citations) in the revised revision. Thank you for your attention to detail!
> >
> > **References:**
> >
> > [1] Gu, Zheng, et al. "Analogist: Out-of-the-box visual in-context learning with image diffusion model." ACM TOG, 2024.
> >
> > [2] Yang, Yifan, et al. "Imagebrush: Learning visual in-context instructions for exemplar-based image manipulation." NeurIPS, 2023.
> >
> > [3] Sundaram, Shobhita, et al. "Personalized Representation from Personalized Generation." ICLR, 2024.
> >
> > [4] Samuel, Dvir, et al. "Where's Waldo: Diffusion Features For Personalized Segmentation and Retrieval." NeurIPS, 2024.

---

### Official Review · Reviewer_HDzZ · 2025-11-01

**Soundness:** 3
**Presentation:** 3
**Contribution:** 3
**Rating:** 6
**Confidence:** 3

**Summary:**

This paper considers the problem of efficiently solving personalized vision tasks which often require an unseen, specific visual transformation. The authors propose a simple training algorithm (PICO) with flow-matching to efficiently fine-tune existing generalist models, and they curate the VisRel dataset which contains diverse, structured examples. The paper also proposes an inference method that uses an attention-based scorer to improve sampling quality. The main empirical results show (both qualitatively and quantitatively) that their method generalizes better than baselines, is more sample-efficient, and works well across a variety of tasks and settings.

**Strengths:**

1. The paper presents several concrete artifacts and ideas that are novel and likely of significance to the community. The most prominent contributions in my view are the VisRel dataset and idea of using an attention-guided seed scorer for inference scaling.
2. I think the greatest strength of the paper is the empirical rigor. The authors do a thorough job of testing various ablations (task generalization, effect of including text, behavior under various axes of scaling), which strengthens the clarity and quality of the core set of results. The ablations build confidence in the significance of their method.
3. In addition, I thought the paper did a good job of presentation, including a nice mixture of both quantitative results as well as corresponding qualitative results (which highlight the benefit of their method). The authors clearly and concisely explain the core parts of their method. I also appreciated the analysis in the appendix on the inference scaling method and the one-page view of the full dataset.
4. Overall, I feel like the quality of the paper is strong, and the paper also includes novel and significant research artifacts and ideas that I believe are of interest to the community.

**Weaknesses:**

1. I have some reservations about the clarity of the paper with respect to how the paper contextualizes prior work in Visual ICL and how their contribution of framing personalized vision fits within this. See Question 1 for more detail.
2. There's a lack of clarity explaining some of the baselines in the main experiments as well as other details of the experiments. This results in some confusion when interpreting the experimental results. See Questions 3-5 below for details. Addressing these questions will improve the clarity and quality of the paper.

**Questions:**

I have various questions about the claims and experimental details. My current rating is a 6, but I'm between a 6 and 8 and I'm willing to increase my score if these are addressed.

1. I'm a bit confused about the contribution claim of "formulating personalized vision as visual in-context learning". Visual ICL as a setting / method (which is typically task-specific) has been around for a few years. Personalized vision is also defined relative to a distribution of downstream tasks, so I'm missing what exactly the contribution is. To be clear, I think the actual artifact of the VisRel dataset is valuable, but I'd appreciate some clarification on why this is a new formulation.

2. Is the task taxonomy used in any way during training? Is the main point to highlight the benefit of task diversity? Also, how exactly are the Semantic complexity and Spatial Locality quantified. In 3.1, they seem to be composed of multiple aspects.

3. I found the main comparison in Table 1 to baselines to be pretty confusing because it was unclear a. which methods were training free b. which methods required more data for training c. which methods were fine-tuned from different models. Adding an additional column classifying this would help a lot for clarity. The table also makes a distinction between large-scale, personalized, generalist methods, but then in Table 2 bottom (labeled large-scale pretrained segmentation models), both large-scale and generalist methods appear. Is the point of Table 2 bottom to show the data efficiency of PICO? At a higher level, this lack of detail on baselines feels more confusing because many of the comparisons in the paper are phrased as "Visual ICL" vs "fine-tuning", but Visual ICL also relies on (a limited amount of) training and some of the generalist methods that you are comparing against are training-based (OmniGen) while others are inference-based (VP). To summarize my questions:

a. Is Table 2 bottom only to show data efficiency? Do you have thoughts on why your method is much more data efficient?

b. What are the "win conditions" that you're comparing methods on? Beyond performance, it seems like the set of all methods you tested have different considerations of training-free, how much data is needed, and which models they fine-tune off of. Can you be more clear about these to highlight your wins and show the comparisons are fair?

c. Do any of the other methods also fine-tune off of FLUX? How much of the wins are coming from this?

4. Why does performance increase between 12 and 48 shots per task in Figure 6A-a? More generally, why is performance not monotonic in data and task scaling?
5. Can the TTC method be applied to models that aren't fine-tuned using PICO and would it help in those cases as well?

---

> ### Author Response · Authors · 2025-11-22
> **(1/2)**
>
> Thank you for your careful reading and insightful comments, and for recognizing the novelty and significance of our contributions, including the VisRel dataset, the attention-guided seed scorer, and the thorough ablations and analysis, as well as the overall clarity of the presentation. We address the questions in detail below; please let us know if there are further comments or concerns.
>
> > **What is actually new in “formulating personalized vision as visual ICL”?**
>
> Prior “personalized vision” work [1, 2, 3] focuses on personalized objects within a single task (e.g., instance segmentation). In this data-scarce scenario, previous methods such as PRPG [1] **use generative priors to synthesize more object-specific data** for training personalized representations per subject, which is very computational expensive, and inflexible.
>
> PICO, on the other hand, **treats the generative model itelf as a visual in-context learner**, enabling it to flexibly adapt to both personalized objects and tasks at test time without additional training. This shift, from training separate personalized models/representation to using a unified diffusion transformer that personalizes via visual ICL, is what we mean by “formulating personalized vision as visual ICL.”
>
> > **Task taxonomy: is it used in training, and what is it for?**
>
> The taxonomy is not used explicitly during training. It is a design and analysis tool that guides us on how to select tasks so that VisRel can cover diverse regions of 2D transformation space (semantic-complexity × spatial-locality), instead of being skewed toward a single task type. Its role is to make task diversity in VisRel more explicit and controllable, and to structure our analysis.
>
> The two axes are conceptual (not quantified), based on the nature of each task:
>
> - "Semantic Complexity: how much semantic understanding the task requires?" For example, low-level tasks (e.g., dehazing, denoising) require little semantic understanding, while tasks like segmetnation or sticker addition require object-/class-level reasoning and thus lie at high end.
>
> - "Spatial Locality: how large the spatial context must be?" Tasks like edge detection or denoising rely mainly on local neighborhoods, while tasks like dehazing or depth estimation require global context and are placed higher on this axis.

---

> > ### Author Response · Authors · 2025-11-22
> > **(2/2)**
> >
> > > **Main comparison (personalized seg.) (Table 1 & Table 2)**
> >
> > Thank you for the detailed questions on the baselines. We clarify our design here.
> >
> > >> **Overall baseline selection and fairness.**
> >
> > Since our work targets a new setting, i.e, evaluating the ability to solve unseen personalized tasks (both object- and task-level), it is quite challenging to find perfectly aligned baselines. We tried our best to ensure fairness by selecting the strongest available baselines and covering a broad range of methods: (1) large-scale, (2) personalized, (3) generalist.
> >
> > For the generalist models, we clarify that all of them are "training-based". VP [4] is described as "inference-based" because it is trained via MAE-style random masking, rather than explicitly in a four-panel, task-relation formulation as others.
> >
> > To avoid confusion and follow your suggestion, **we have expanded the generalist methods in Table 2 (bottom) with detailed setups in the revised version.**
> >
> > >> **a. Table 2 bottom & data efficiency.**
> >
> > After revision, the bottom of Table 2 is used to show:
> > - training data efficiecny and the ratio/balanced of the segmentation data vs. all data.
> > - training paradigm, how it is trained, which loss it used, and which pretrained model it builds on.
> >
> > We attribute PICO’s data efficiency mainly to:
> > - the strong generative prior, yielding rich, reusable visual representations.
> > - the DiT architecture, which naturally support in-context learning, making it efficient to learn exemplar-query relations.
> > - the inter- and intra-task diversity in VisRel, which encourages the model to learn general “visual relations” rather than overfitting to a single task.
> >
> > >> **b. Win conditions.**
> >
> > For the personalized image segmentation, our comparisons are guided by three criteria, with a focus on the personalized vision setting:
> > 1. Performance on unseen personalized tasks.
> > 2. Practicality and flexibility of personalization. (e.g., whether the method can adapt to new objects and new tasks easily).
> > 3. Data efficiency and training efficiency.
> >   This is relatively minor compared to first two criteria, but important in personalized settings, where data is always limited and large-scale labeled datasets are often infeasible to obtain.
> >
> >
> > >> **c. Dependence on FLUX.**
> >
> > None of the baselines are finetuned from FLUX. They use their own backbones (diffusion U-Nets, ViTs, LLaMA-style transformers, etc.).
> >
> > The FLUX DiT prior certainly provides a strong starting point, but our ablation studies (e.g., varying VisRel task diversity, droping the text, etc.) isolate contributions within our method and attribute performance gain to our method.
> >
> > Moreover, contributions such as leveraging task diversity for better generalization and the heuristic seed-scorer for test-time scaling are orthogonal to the specific architecture and, in principle, could also be applied to other backbones.
> >
> >
> > > **Non-monotonic behavior in Figure.6**
> >
> > In Figure 6, every point is trained under the same budget (fixed steps and model size), while the amount of data and the number of tasks vary. As we add more data/tasks under a fixed budget, the effective updates per example and per task decrease, so performance is not guaranteed to be strictly monotonic.  The figure is intended to illustrate overall trends and trade-offs, rather than to claim perfectly monotonic scaling curves.
> >
> >
> > > **Applicability of TTS beyond PICO.**
> >
> > Yes. TTS (our attention-guided seed selection) is training-free and only uses early cross-attention statistics between the target region (BR) and its visual context (A, A′, B).  In principle, the same scoring rule can be plugged into any DiT- or diffusion-based model that exposes cross-attention maps with spatial visual conditions (e.g., in-context editing, exemplar-based manipulation).
> >
> >
> > **References:**
> >
> > [1] Sundaram, Shobhita, et al. "Personalized Representation from Personalized Generation." ICLR 2024.
> >
> > [2] Nguyen, Thao, et al. "Yo'Chameleon: Personalized Vision and Language Generation." CVPR, 2025.
> >
> > [3] Samuel, Dvir, et al. "Where's Waldo: Diffusion Features For Personalized Segmentation and Retrieval." NeurIPS, 2024.
> >
> > [4] Bar, Amir, et al. "Visual prompting via image inpainting." NeurIPS, 2022.

---

### Official Review · Reviewer_xMbv · 2025-11-01

**Soundness:** 3
**Presentation:** 4
**Contribution:** 3
**Rating:** 6
**Confidence:** 5

**Summary:**

This paper introduces a concept called the Personalized In-Context Operator (PICO), which allows personalized visual tasks to be carried out through visual in-context learning with a diffusion transformer. The framework is a four-panel framework where single annotated exemplars are presented to the model, the PICO infers the underlying transformations and apply it to the new inputs. In addition to the operator, the paper proposes VisRel, a diverse tuning dataset used to train the model for this task. The work benchmarks their method against an extensive set of experiments, demonstrating that PICO surpasses various baselines and is able to flexibly adapt to various user-defined tasks, spanning both recognition and generation.

**Strengths:**

- **Flexibility with personalization provided by the ICL framework:** The ICL format of PICO allows this method to be flexible in adapting to new user-defined task without retraining. This is a strong step towards open-ended user-driven visual inference.
- **Robustness and stability:** The authors propose inference scaling seed selection method to prevent diverging seeds and identifies stable seeds using early cross-attention routing
- **Diverse Task Generalization:** In addition to standardized personalized detection / segmentation tasks, the authors also test PICO in various flexible tasks that take advantage of the ICL framework - such as composite taks, spatially constrained tasks, and semantic-conditional tasks. The performance shows PICO
- **Data and Compute Efficiency:** It enables adaptation to new tasks without costly finetuning or model specialization

**Weaknesses:**

- **Generalization limits:** It is noted that PICO is less reliable on entirely novel task types “outside the trained visual-relation space”. This limits its practical applicability for highly novel or out-of-distribution (OOD) tasks, which are key for real-world personalization
- **Limited demonstration richness:** The current framework only supports one-shot demonstrations, which constrains the expressive power of the provided exemplars. A multi-shot setting could better capture complex or composite transformations.
- **Evaluation coverage:** While the paper presents strong results on diverse benchmarks, most tasks are represented within VisRel or are closely related to it. More systematic evaluation on unseen OOD transformations would better validate the claimed generalization ability.

**Questions:**

- **Generalization:** How does PICO perform on entirely novel tasks outside of the VisRel taxonomy —  most of the benchmarked datasets were included in VisRel in one form or another (e.g. inpainting, edge detection, segmentation, etc.).  For OOD tasks, is there difference in task performance with respect to other baselines? Could you show some qualitative examples of OOD tasks as well or show benchmarks?
- **Multi-shot ICL:** Could PICO be extended to handle multi-shot or sequential visual demonstrations? Given the reliance on a four-panel grid, how would you handle richer or more structured visual contexts (e.g., videos or multiple exemplars)? Do you expect consistent performance gains with additional exemplars?
- **Task diversity vs. capacity trade-off:** Figure 6 shows that task diversity drives generalization, but increasing the number of tasks hurts in-domain performance. What do you think is a bottleneck to this phenomenon, is it a computational constraint or limitations in the unified visual-relation space itself?

---

> ### Author Response · Authors · 2025-11-22
>
> Thank you for your careful reading and thoughtful review, and for recognizing the flexibility and efficency of PICO, the stabilizing effect of our seed-selection strategy, and its performance on diverse personalized tasks. We address the questions in detail below; please let us know if there are further comments or concerns.
>
> > **Generalization to entirely novel tasks outside the VisRel taxonomy**
>
> Our work targets **personalized vision** in two senses of “novel”:
> 1. **Novel instances**: new objects not seen in VisRel (*e.g.*, personalized image segmentation in Sec. 4.2).
> 2. **Novel task definitions within/close to the VisRel relation space**: user-defined composite / constrained tasks (interpolated) created only at test time (Sec. 4.3).
>
> As you correctly point out, there are task types whose underlying relations are **not** covered by the current VisRel space, such as **abstract reasoning problems** (e.g., ARC-style visual puzzles [1]).
>
> We add a small qualitative study on ARC-like tasks in the revised version (please see Figure. 9), formulating them as one-shot visual analogies.  PICO can produce outputs that are visually resembled but are logically incorrect. Extending the visual-relation space toward such reasoning-heavy tasks is an interesting direction for future work.
>
> > **Multi-shot ICL**
>
> Yes, PICO can be extended to multi-shot or sequential visual demonstrations. The diffusion transformer (DiT) processed inputs as sequences of tokens rather than fixed channels (Unet).
>
> Our four-panel is a design choice for minimal and efficiency. To handle richer contexts, we can simply extend the token sequence to include multiple examplar pairs (e.g., a 3x2 grid).
> We expect consistent performance gains with additional examplars, especially for more complex or ambiguous tasks, because richer visual demonstrations help disambiguate the intended transformation.
>
> > **Task diversity vs capacity trade-off**
>
> We view the drop in in-domain performance as diversity increases (Figure 6) primarily as a **capacity/optimization** issue rather than a limitation of the unified visual-relation space.
>
> In our experiments, we kept the model size and training budget fixed. As we add more task types, the “gradient budget” per task decreases, and the model allocates its finite capacity to more diverse relations. This encourages more generalized features (which improves cross-task generalization, e.g., personalization) but slightly hurts specialized in-domain metrics.
>
> We see this as a standard multi-task trade-off. Scaling the model size alongside task diversity and the shots per task would likely reduce this drop, and we leave a systematic study of this scaling behavior to future work.
>
> **References:**
>
> [1] Li, Wenhao, et al. "Tackling the abstraction and reasoning corpus with vision transformers: the importance of 2d representation, positions, and objects." arXiv preprint arXiv:2410.06405 (2024).

---

### Official Review · Reviewer_V1LZ · 2025-11-02

**Soundness:** 3
**Presentation:** 3
**Contribution:** 2
**Rating:** 4
**Confidence:** 3

**Summary:**

This paper introduces Personalized In-context Operator-  PICO, a framework that applies visual in-context learning to personalized vision tasks. The authors reformulate personalized vision as a visual ICL problem, where a single input-output exemplar defines the task through a four-panel format。 To support this approach, they construct VisRel, a compact training dataset of 27 diverse visual tasks organized by semantic complexity and spatial locality. The method builds upon pretrained diffusion transformers fine-tuned with LoRA, and introduces an attention-guided seed scorer for test-time scaling. Experiments demonstrate that PICO achieves competitive performance on personalized segmentation benchmarks while using significantly less labeled data than baseline methods, and shows the ability to handle user-defined tasks at test time without additional fine-tuning.

**Strengths:**

- The proposed method demonstrates  improvements over baseline approaches across multiple benchmarks. The attention-guided seed selection strategy is empirically effective.
- The framework successfully handles a diverse range of personalized vision tasks.
- The paper is well-organized and mostly clearly written.

**Weaknesses:**

Please see questions below for details.
- Critical details regarding the evaluation setup are missing or unclear, which hinders a thorough assessment of the method's effectiveness and generalization capabilities.
- While the authors emphasize that VisRel is compact (315 samples across 27 tasks), the extent to which the method's success depends on having explicit training coverage of test task types remains unclear. This is particularly important given the paper's central claim about generalization to novel, user-defined tasks.

**Questions:**

- Based on Table 5, most evaluation tasks appear to have corresponding task types in the VisRel training set. Could the authors clarify which evaluation tasks represent truly novel task types not covered during training? Without this clarification, it is difficult to distinguish between interpolation within the trained relation space versus genuine zero-shot generalization to unseen tasks.
- In Figure 6 and the discussion around line 450, the distinction between "seen" and "unseen" tasks needs clearer definition. Specifically, what constitutes a "seen" versus "unseen" task in this experiment, e.g., novel objects for completely novel tasks?
- In Table 6, and Appendix C.2, why were only 3 tasks removed per category rather than entire task families? Can the authors provide ablations where entire task categories are removed and evaluate on held-out task categories to demonstrate true cross-task generalization?
- What is the computational overhead of the proposed TTS method? Specifically: 1) How does inference time scale with the number of candidate seeds |S|? 2) What is the trade-off between computational budget and performance improvement? 3) Are there diminishing returns beyond a certain number of seeds?

---

> ### Author Response · Authors · 2025-11-22
> **(1/2)**
>
> Thank you for your constructive feedback and for acknowledging the improvements, effectiveness on personalized vision tasks, and overall organization of the paper. We address the questions in detail below; please let us know if there are further comments or concerns.
>
> > _**Which evaluation tasks are truly novel vs interpolation within VisRel?**_
>
> We evaluate PICO in two personalized vision settings:
>
> - new objects/instances (Sec. 4.2): personalized object segmentation
> - novel new tasks (Sec. 4.3): tasks whose specification differs from conventional setups and are **not covered during training**, including composite tasks, spatially constrained tasks, and semantic-conditional tasks. For example, we do not train on watermark removal. PICO solves it by composing the logic of inpainting (restoration). Whether it is a large central logo (Fig. 4a) or small tiled marks(Fig. 4b), the model infers the "remove and restore" instruction from the exemplar pair.
>
> Addressing "interpolation": You are corrent that this is "interpolation within the trained relation space". However, we argue this kind of compositional generalization is **non-trivial and practically desirable**.
> Existing visual ICL methods often fail to compose these primitives effectivesly [1, 2, 3], whereas PICO can synthesize these complex “visual instructions” directly at test time.
>
> > **“Seen” vs “unseen” in Figure 6**
>
> In Figure 6, “seen” and “unseen” refer to task types (see line 455-456). The seen task uses surface normal estimation, which is included during training.  The unseen task uses personalized image segmentation, which is not included during training and also involves novel objects.
>
> > **Why remove only 3 tasks per category in Table 6? Entire task families?**
>
> Our original choice to remove 3 tasks per category was to probe sensitivity to regions of the taxonomy **without drastically changing overall task diversity and data volume**.
> Removing an entire would family at once (i) remove that relation type, (ii) heavily reduce total diversity, and (ii) reallocates much more optimization budget to the remaining tasks, which makes the results harder to interpret.
>
> Following your suggestion, we additionally ran full-family removal experiments (removing all physical, low-level, generative, or semantic tasks), and then evaluate on personalized image segmenation. The results are:
>
>
> | Method   |              |     PerSeg      |              |              |    DOGS       |              |              |    PODS       |              |              |    PerMIS        |              |
> |----------|--------------|----------------|--------------|--------------|---------------|--------------|--------------|---------------|--------------|--------------|------------------|--------------|
> |          | mIOU ↑         | bIOU ↑           | F1 ↑         | mIOU ↑         | bIOU ↑           | F1 ↑         | mIOU ↑         | bIOU ↑           | F1 ↑         | mIOU ↑         | bIOU ↑           | F1 ↑         |
> | PICO | 90.97    | 76.13       | 62.82     | 71.02    | 54.71      | 49.84    | 68.72     | 60.26      | 44.88     | 49.52     | 33.63        | 14.90     |
> | Remove physical | **94.75**     | **78.90**    | **98.43**    |  75.51    |  60.66  |   61.60   |  63.94  |   59.84   |  47.40  |    **58.89**     |    38.33     |   **22.02**   |
> | Remove low-level   | 94.04      | 77.60     | 64.66     |   79.09   |  61.11  |    65.21  |  73.70  |   **66.97**   |  51.11  |    48.25     |    31.01     |   13.57   |
> | Remove Generative  | 93.74       | 77.92      | 97.36    |   **82.30**   |  **63.07**  |   **66.30**   |  72.47  |  65.64    |  **52.40**  |   55.76      |   **38.90**      |   17.26   |
> | Remove Semantic |   68.41   |    51.56     |   70.07   |   44.40   |   29.87   |  23.16  |   49.79   |  42.68   |  31.52  |    29.19     |    16.55     |   5.71   |
>
> Two observations:
> - Removing semantic tasks causes a large drop.
> - Removing non-semantic families does not hurt performance and improve some metrics. We attribute these gains to a redistribution of capacity and training steps toward the remaining segmentation-like and semantic tasks, rather than these families being “harmful” to generalization.
> We will include these full-family removal results and a brief discussion of their impact on test-time task generalization (Sec. 4.4) in the final version.

---

> > ### Author Response · Authors · 2025-11-22
> > **(2/2)**
> >
> > > **Computational overhead and trade-offs of the TTS method**
> >
> > Thank you for the suggestion. we add an ablation on personalized image segmentation (please see revised Figure. 8), where we vary the seed pool size $|S|\in\{1,10,20,50\}$ and report both performance and inference time.
> >
> > Inference time grows approximately linearly with $|S|$ (e.g., 40.6s at $|S|{=}1$, 193.8s at $|S|{=}10$, 353.5s at $|S|{=}20$), while performance improves clearly from $|S|{=}1$ to $10$ but only marginally beyond $|S|{=}20$, showing a clear diminishing return.
> >
> > **References:**
> >
> > [1] Bar, Amir, et al. "Visual prompting via image inpainting." NeurIPS, 2022.
> >
> > [2] Wang, Xinlong, et al. "Images speak in images: A generalist painter for in-context visual learning." CVPR, 2023.
> >
> > [3] Bai, Yutong, et al. "Sequential modeling enables scalable learning for large vision models." CVPR, 2024.

---

### Author Response · Authors · 2025-12-03
**Global Response**

We thank all of the reviewers for their helpful and insightful feedback, and we are grateful to the AC for the additional effort and careful oversight in handling the review process. We are glad that they found:

- The paper to be well-organized and clearly written. [`V1LZ`, `xMbv`, `HDzZ`, `H62u`]

- PICO to be a flexible and data-efficient visual ICL method with strong performance on user-defined personalized vision tasks at test time. [`V1LZ`, `xMbv`, `H62u`]

- The attention-guided seed selection strategy to be effective and helpful for improving robustness and stability during inference. [`V1LZ`, `xMbv`, `HDzZ`, `H62u`]

- The experiments to be thorough and comprehensive, with diverse task coverage. [`V1LZ`, `xMbv`, `HDzZ`, `H62u`, `Cvqg`]

- The VisRel dataset and related artifacts to be useful and of interest to the community. [`HDzZ`]

- The emphasis on task variety (rather than size) to be a good direction for visual ICL research [`H62u`]

We highlight the two most common concerns below (and answer individual questions in detail in reviewer-specific responses). Updates are **blue** in our revised PDF.

> **What is actually new beyond prior visual ICL / personalization work [`HDzZ`, `H62u`, `Cvqg`]**

Prior “personalized vision” work [1, 2, 3] focuses on personalized objects within a single task (e.g., instance segmentation). In data-scarce settings, methods such as PRPG [1] **use generative priors to synthesize more object-specific data** for training personalized representations per subject, which is very computational expensive, and inflexible.

Existing visual ICL methods share the similar "example-based" interface, but are either (i) desigend mainly for semantic-driven image manipulation [4, 5, 6] and cannot handle perception tasks, or (ii) predefined multi-task models [7, 8] trained on large-scale classic CV tasks, and lack the in-context generalization needed in personalized setting. **In other words, they are not “personalized” in the sense we target, because the space of transformations they can adapt to is fixed by training.**

PICO instead **treats the generative model itelf as a visual in-context learner**, together with the **diverse VisRel relation space**, enables one-shot adaptation to both new objects and new user-defined tasks at test time without additional training. Our results show this formulation works really well for "personalized vision" in ways not demonstrated in prior work, helping to close the gap between personalized vision and visual ICL.

>  **What “generalization” we actually claim [`V1LZ`, `Cvqg`]**

We evaluate PICO in two personalized vision settings:

- **New objects/instances** (Sec. 4.2): personalized segmentation on objects that **never appear** in training.
- **Novel new tasks** (Sec. 4.3): tasks whose specification differs from conventional setups and are **not covered** during training, including composite tasks, spatially constrained tasks, and semantic-conditional tasks.

As `V1LZ` notes (and as we already state around lines 168–169), this test-time task generalization is best viewed as interpolations and composition within our diversity-driven visual relation space. Our point is that this kind of compositional generalization is non-trivial and practically important, and existing visual ICL methods fail to do so.

At the same time, as we acknowledge in our limitations and in response to `xMbv`'s question on generalization limits, there are task types whose underlying relations are not covered by the current VisRel space, such as abstract reasoning problems (e.g., ARC-style visual puzzles). In the revision, we add a qualitative study on ARC-like tasks (Figure. 9): PICO can produce outputs that are visually plausible but logically incorrect. Extending the visual-relation space toward such reasoning-heavy tasks is an interesting direction for future work, not as part of our current generalization claims.

---

> ### Author Response · Authors · 2025-12-03
>
> **References:**
>
> [1] Sundaram, Shobhita, et al. "Personalized Representation from Personalized Generation." ICLR 2024.
>
> [2] Nguyen, Thao, et al. "Yo'Chameleon: Personalized Vision and Language Generation." CVPR, 2025.
>
> [3] Samuel, Dvir, et al. "Where's Waldo: Diffusion Features For Personalized Segmentation and Retrieval." NeurIPS, 2024.
>
> [4] Gu, Zheng, et al. "Analogist: Out-of-the-box visual in-context learning with image diffusion model." ACM TOG, 2024.
>
> [5] Lai, Bolin, et al. "Unleashing in-context learning of autoregressive models for few-shot image manipulation." CVPR, 2025.
>
> [6] Yang, Yifan, et al. "Imagebrush: Learning visual in-context instructions for exemplar-based image manipulation." NeurIPS, 2023.
>
> [7] Bar, Amir, et al. "Visual prompting via image inpainting." NeurIPS, 2022.
>
> [8] Wang, Xinlong, et al. "Images speak in images: A generalist painter for in-context visual learning." CVPR, 2023.

---

### Meta-Review · Area_Chair_ZZGV · 2026-01-06

**Summary:**

This work introduces personalized vision generation via visual in-context learning. Specifically, the personalized task can be defined by an annotated exemplar and then given new inputs, the model will generate the appropriate outputs with a fine-tuned diffusion model. In addition, a seed scorer is developed to further improve the performance when inference. Experiments on diverse tasks show that the proposed method can handle those tasks well with a limited number of training data.

**Reviewer Concerns:**

A major concern shared by multiple reviewers is the generalization of the proposed algorithm (Reviewer V1LZ, xMbv and Cvqg). While the motivation of this work is to handle the arbitrary tasks defined by users, the rebuttal confirms that the current version is mainly for tasks partially covered by training set, which limits the application. Another concern is about the novelty compared to existing algorithms (Reviewer HDzZ, H62u, Cvqg). While the target tasks can be different, the technique proposed in this work is similar to existing work. Besides, Reviewer HDzZ and Cvqg have the concern about the strong backbone adopted in the proposed method, i.e., FLUX, compared to baselines. Reviewer V1LZ has the concern about the overhead of the proposed TTS method and the rebuttal provides a detailed discussion.

**Reviewer Scores:**

This work received mixed scores as 4,6,6,4,4 before rebuttal. While the work aims to address an interesting problem, the rebuttal shows that the current result may not be sufficient, especially for the generalization to novel tasks and appropriate discussion/comparison to existing work. Therefore, the initial scores can be retained after rebuttal. It will be a strong submission by including suggestions from reviewers in the next revision.

---

### Decision · Program_Chairs · 2026-01-26

Reject